# Hacking the Immune Response to Solid Tumors: Harnessing the Anti-Cancer Capacities of Oncolytic Bacteria

**DOI:** 10.3390/pharmaceutics15072004

**Published:** 2023-07-21

**Authors:** Jason M. Roe, Kevin Seely, Caleb J. Bussard, Emily Eischen Martin, Elizabeth G. Mouw, Kenneth W. Bayles, Michael A. Hollingsworth, Amanda E. Brooks, Kaitlin M. Dailey

**Affiliations:** 1College of Osteopathic Medicine, Rocky Vista University, Ivins, UT 84738, USA; 2College of Osteopathic Medicine, Rocky Vista University, Parker, CO 80130, USA; 3Department of Pathology and Microbiology, University of Nebraska Medical Center, Omaha, NE 68198, USA; 4Eppley Institute for Cancer Research, University of Nebraska Medical Center, Omaha, NE 68198, USA; 5Office of Research & Scholarly Activity, Rocky Vista University, Ivins, UT 84738, USA

**Keywords:** oncolytic bacteria, bacterial-mediated cancer therapeutics, host–pathogen interaction, immune response, synthetic biology, programmable medicine, oncology, engineered bacterial therapeutics, tumor therapy, solid tumor, microbiome

## Abstract

Oncolytic bacteria are a classification of bacteria with a natural ability to specifically target solid tumors and, in the process, stimulate a potent immune response. Currently, these include species of *Klebsiella*, *Listeria*, *Mycobacteria*, *Streptococcus*/*Serratia* (Coley’s Toxin), *Proteus*, *Salmonella*, and *Clostridium*. Advancements in techniques and methodology, including genetic engineering, create opportunities to “hijack” typical host–pathogen interactions and subsequently harness oncolytic capacities. Engineering, sometimes termed “domestication”, of oncolytic bacterial species is especially beneficial when solid tumors are inaccessible or metastasize early in development. This review examines reported oncolytic bacteria–host immune interactions and details the known mechanisms of these interactions to the protein level. A synopsis of the presented membrane surface molecules that elicit particularly promising oncolytic capacities is paired with the stimulated localized and systemic immunogenic effects. In addition, oncolytic bacterial progression toward clinical translation through engineering efforts are discussed, with thorough attention given to strains that have accomplished Phase III clinical trial initiation. In addition to therapeutic mitigation after the tumor has formed, some bacterial species, referred to as “prophylactic”, may even be able to prevent or “derail” tumor formation through anti-inflammatory capabilities. These promising species and their particularly favorable characteristics are summarized as well. A complete understanding of the bacteria–host interaction will likely be necessary to assess anti-cancer capacities and unlock the full cancer therapeutic potential of oncolytic bacteria.

## 1. Introduction and Background

A subset of bacteria have demonstrated an ability to specifically target and lyse cancer cells, garnering the designation ‘oncolytic’, or cancer lysing [1]. Oncolytic bacteria offer a more selectively targeted approach for cancer therapeutics, with fewer side effects than current modalities, such as chemo- or radiotherapy, in addition to several novel advantages. Currently, these include species of *Klebsiella*, *Listeria*, *Mycobacteria*, *Streptococcus/Serratia* (Coley’s Toxin), *Proteus*, *Salmonella*, and *Clostridium* (Figure 1). Interest in oncolytic bacteria initially dwindled due to an inability to effectively limit systemic toxicity inherent to bacterial administration that often led to sepsis. Recent advances in cellular engineering, directed evolution, and synthetic biology have generated a new wave of interest. With the advent of new technologies, innate anti-cancer capacities of oncolytic bacteria can be both harnessed and amplified through bioengineering, specifically through new genetic engineering methodology such as CRISPR/Cas systems. In theory, oncolytic bacteria could expand or immunologically “hijack” the body’s immune response to elicit a tumorolytic response [2].

The characteristically harsh microenvironments of solid tumors present advantageous niches for oncolytic bacteria localization and colonization, and thus subsequent tumor destruction paired with potent immune restimulation. Solid tumors are typically defined as neoplasms that arise from solid tissues such as bones, muscles, and organs [3]. According to the World Health Organization (WHO), cancer is the second leading cause of death globally, with approximately 50% of these deaths being due to solid tumors [4]—highlighting the need for more effective and alternative treatments. Despite significant advances in understanding solid tumors, the treatment of these harsh microenvironments continues to pose unique challenges for canonical drug development [5]. This literature review highlights potential strategies to harness the natural anti-cancer capacities of oncolytic bacteria and discusses specific characteristics that will prove critical for clinical translation as therapeutics (e.g., secretions, surface molecules, and host interactions). The purpose of this manuscript is to provide a unique, unpublished perspective in that it specifically pairs the literature detailing oncolytic bacteria surface characterization with the reported host–immune bidirectional interactions in pathogenic contexts to develop novel oncotherapeutics and live biologic therapies with the ultimate goal of clinical translation.

Surgery is often the first line of treatment for solid tumors and obviously involves the physical removal of the cancerous tissue [5]. While surgery can be highly effective in the treatment of early-stage cancers, its efficacy decreases with disease progression and can be quite limited in advanced cancer stages [6]. Furthermore, many tumors are not accessible using modern surgical techniques, or have metastasized to multiple locations prior to detection. Surgery with adjuvant or neoadjuvant radiation and/or chemotherapy is currently the first line of treatment for solid tumors. Other biologic adjuvants are also known to modulate the immune response by targeting neoplasms as reviewed elsewhere [7,8]. Radiation therapy employs high-energy radiation to destroy tumor cells [9], and is often administered in combination with other treatments, such as chemotherapy. Chemotherapy administers drugs to destroy cancer cells. When administered intravenously or orally, these therapeutics can have significant side effects (e.g., nausea, vomiting, hair loss, and decreased immune function) [10]. Immunotherapy, or the technology behind individualized or precision medicine, is becoming increasingly implemented—though costly and time consuming to formulate. Immunotherapy modulates the immune response to tumorigenic cells, intensifying the natural response [11]. While highly effective in some patients, minimal effects have been observed for others (e.g., pancreatic cancer) [11]. The efficacy of immunotherapy seems to vary widely and likely depends on intrinsic tumor characteristics that do not affect oncolytic bacteria mediated treatments [11].

Oncolytic bacteria demonstrate several advantages over canonical therapeutics, including overcoming drug resistance by targeting chemotherapy-resistant cancer stem cells [12]. In addition, oncolytic bacteria are thought to reduce the risk of tumor recurrence by breaking down the physical and biological barriers that can shield cancerous cells from chemotherapy and radiation penetration [13]. Oncolytic bacteria can stimulate immunomodulation in addition to direct lytic effects by causing a targeted host immune response directed at tumorigenic cells. The purpose of this review is to pair the published literature regarding bacterial cell surface, metabolism, and interaction with the host in pathogenic infections with developments that have been made in efforts toward clinical translation. For each oncolytic bacterial species, a discussion of the basic microbiology and genome characteristics is paired with important cell surface molecules, metabolism, byproducts, and secretions to provide an adequate context for further development using modern techniques such as genetic engineering to harness these species as the next generation of biologic therapeutics.

### 1.1. Essential Components of Host–Pathogen Interactions

The composition of the cell and spore surfaces of oncolytic bacteria is of critical importance to confer added oncolytic capacities, as is a fundamental knowledge of how these species interact and respond to environmental context—particularly those that replicate solid tumor microenvironments. It has been well detailed that microorganisms maintain a cell membrane optimal fluidity, adjusting composition to allow for optimal viscosity and ionic diffusion for enzymatic activity. These changes occur through altered membrane constituent synthesis as governed by environmental context. Each constituent contributes uniquely to the membrane composition, and thus minor alterations in composition may have far reaching physiological consequences. This review pairs the literature regarding the known surface proteins and cell secretions with their elicited immunogenic effects to further the field of oncolytic bacteria-mediated cancer therapeutics. Thorough discussion, including figures, is specifically focused on strains that have accomplished Phase III clinical trial initiation (*Klebsiella pneumoniae*, *Listeria monocytogenes*, *Salmonella typhimurium*, and *Clostridium novyi*-NT) as representatives of their respective species.

There are several essential components of the host–pathogen interaction that play a critical role in the activation of the innate immune system worth noting prior to an in-depth discussion of specific oncolytic species/strain characteristics. These molecular structures are nearly ubiquitous, found in a wide range of pathogens, or are specific to a particular group of microorganisms (e.g., Gram-positive vs. Gram-negative). Important among these components for this discussion of harnessing the anti-cancer capacities of oncolytic bacteria are pathogen-associated molecular patterns (PAMPs) [14], damage-associated molecular patterns (DAMPs) [15], and neutrophil extracellular traps (NETs) [16].

PAMPs and DAMPs are evolutionarily conserved protein motifs common enough to be readily recognized by the host’s innate immune system [17]. PAMPs are expressed on pathogenic microorganism membranes and when detected act as indicators of pathogen presence [17]. DAMPs are endogenous motifs present on proteins released from damaged cells [17]. The recognition of PAMPs and DAMPs triggers a series of events that ultimately lead to the general production of pro-inflammatory cytokines, such as interleukin-8 (IL-8), and stimulation of the adaptive immune response [17]. The innate immune responses are crucial for routine pathogenic defenses and the maintenance of tissue homeostasis, but excessive activation can lead to chronic inflammation and thus disease development.

Lipopolysaccharide (LPS) is a common PAMP located in the outer membrane of Gram-negative bacteria [18]. LPS plays a significant role in the host–pathogen immune response as a potent activator—typically through interactions with Toll-like receptor 4 (TLR-4), which is present on macrophage, dendritic cell, and B cell surfaces. Binding of LPS to TLR-4 triggers a signal cascade of proinflammatory cytokines such as interleukin-1 (IL-1), interleukin-6 (IL-6), and tumor necrosis factor-alpha (TNF-α) [19] (Figure 2). In turn, elevated cytokine levels recruit immune cells to the site of infection, stimulate the production of antimicrobial peptides, and promote the development of an adaptive immune response. However, excessive or uncontrolled activation of the immune system by LPS can also contribute to septic shock, a life-threatening condition characterized by systemic inflammation and organ dysfunction [16,19].

Lipoteichoic acid (LTA) is a PAMP specific to the surface of Gram-positive bacteria [20,21]. In a similar mechanism to LPS, LTA activates macrophages and dendritic cells to produce the pro-inflammatory cytokines IL-1 and TNF-α [22] (Figure 2). LTA is recognized and bound by TLR-2 which then undergoes a conformational change and heterodimerization with other Toll-like receptors [23,24]. This triggers an intracellular signaling pathway using various adaptor proteins to activate IL-1 and TNF receptor kinases [23,24]. These kinases allow for NF-κB to be activated which results in activation of gene transcription for cytokines that result in inflammation and immune cell chemotaxis to the tumor cells [23,24,25]. LTA can also induce the secretion of antimicrobial peptides and activate the complement system [26], which plays a crucial role in the elimination of invading pathogens (Figure 2). Moreover, LTA typically promotes the adaptive immune response by enhancing T cell antigen presentation [27]. However, like LPS, excessive or uncontrolled activation of the immune system by LTA leads to damaging inflammation and subsequent disease states [28], including cardiovascular disease [29,30], atherosclerosis [31], and rheumatoid arthritis [32].

DAMPs include molecules such as HMGB1 (high-mobility group box 1 protein) [33], S100 proteins [34], and ATP (adenosine triphosphate) [35], extracellular cold-inducible RNA-binding proteins [36], histones [37], heat shock proteins [38], extracellular RNAs, and cell-free DNA [39]. The release of DAMPs from damaged cells triggers an innate immune response, leading to the activation of immune cells and the release of pro-inflammatory cytokines. Pattern recognition receptors (PRRs), found on various innate immune cells, are responsible for recognizing DAMPs [40] such as Toll-like receptors (TLRs) [41], NOD-like receptors (NLRs), [42] and RIG-I-like receptors (RLRs) [43]. Upon DAMP recognition, receptors activate signaling cascades that ultimately result in the production of pro-inflammatory cytokines such as IL-6 and IL-8 and the recruitment of immune cells to the site of damage, facilitating tissue repair and immune defense (Figure 2).

Neutrophil extracellular traps (NETs) are also important to consider for developing oncolytic bacteria. NETs are web-like structures composed of DNA, histones, and antimicrobial peptides that are released by neutrophils to trap and kill pathogens [44]. NETs play a crucial role in the host–pathogen immune response [45] and serve to physically entangle and immobilize pathogens, preventing spread and enabling elimination via phagocytosis [44,45]. Further, NETs contain antimicrobial proteins and enzymes that directly kill or degrade invading microbes [46]. However, excessive or uncontrolled formation of NETs can also contribute to tissue damage and inflammation in various diseases, including sepsis and autoimmune disorders. Therefore, the regulation of NET formation is essential for a balanced and effective immune response and must be accounted for when considering oncolytic bacteria application.

The induction of Th1 cytokines and stimulation of a T cell response causes an increased inflammatory response in the tumor microenvironment leading to a decrease in tumor growth. TNF-α can increase the permeability of the vascular lining of tumor cells, allowing drug permeation into tumor cells as well as infiltration of blood and immune cells leading to hemorrhagic necrosis [47]. TNF-α leads to activation of a JNK signaling pathway that contributes to tumor cell death [48,49] (Figure 2). Unfortunately, TNF-α can also inhibit the anti-tumor response and decrease the visibility of tumor cells to surrounding immune cells [48]. IL-6 acts through JAK-STAT signaling pathways, but the effects are dependent on the cells present and the specific context of the tumor microenvironment [50]. According to some studies, IL-6 can have indirect anti-tumor activity through Bcl-2 and immunoglobulin production, but also can stimulate cytokine secretion within the tumor and limit the effectiveness of immunotherapy [50,51]. Nitric oxide can cause DNA damage through production of damaging metabolites as well as causing cell cycle arrest leading to cellular failure to repair DNA damage [52,53]. The damage caused by nitric oxide (NO), as well as activation of signaling networks, can induce apoptosis of tumor cells by binding of death ligands to cell surface receptors and release of cytochrome c [52,53,54]. However, production of nitric oxide can also have pro-tumor effects through creating a hypoxic environment that leads to tumor angiogenesis, so the use of nitric oxide alone to treat tumor cells must be studied further [52,55].

Oncolytic bacteria are not exempt from canonical PAMP, DAMP, and NET pathway stimulation, and these factors must be a consideration in development toward clinical translation (Figure 2). However, the responses stimulated by oncolytic bacteria tumor localization and colonization are thought to not only recruit the innate immune system to target oncolytic bacteria, but in the process, to cause the release of tumor neoantigens that effectively retrain localized immune responses (Figure 2). Ultimately, this creates an environment that is hostile to cancer cells while promoting the activation of immune cells that can attack the tumor as previously reviewed [56]. *Clostridium novyi*-NT, for example, is theorized to utilize LPS to stimulate the innate immune response [57]. In this context, LPS is hypothesized to trigger the release of pro-inflammatory cytokines, leading to the activation of immune cells and the recruitment of immune cells to the site of the tumor, as previously reviewed [57]. A detailed discussion on individual oncolytic bacterial species and their interactions with the aforementioned components follows in their respective sections below. The ability of oncolytic bacteria to utilize PAMPs, DAMPs, and NETs to stimulate the innate immune system provides a promising new approach to the treatment of solid tumors. These bacteria can selectively target and destroy cancer cells while promoting the activation of immune cells that can attack solid tumors.

Gram-positive bacteria are thought to have five major types of surface proteins: (1) proteins anchored to the cytoplasmic membrane by hydrophobic transmembrane domains, (2) lipoproteins covalently attached by the N-terminus to long chain fatty acids of the cytoplasmic membrane, (3) proteins binding to components of the cell wall, (4) proteins attached to the cell surface by S-layer homology domains (SLH), and (5) proteins covalently anchored to the cell wall possessing an LPXTG motif [58]. In contrast, Gram-negative bacteria have two membranes that surface proteins must span to achieve extracellular presentation [59]. This second membrane provides a layer of protection for single celled organisms but can also inhibit nutrient and waste diffusion [59]. In this circumstance most proteins that achieve export across the inner membrane utilize the general sec secretion pathway or the twin arginase translocation (tat) pathway [58]. Once in the periplasm, integral outer membrane proteins (OMPs) are either spontaneously inserted into the outer envelope or require the use of specialized machinery [58]. Periplasmic precursor peptides sometimes assemble into complex structures such as flagella, pili, or S-layers [58].

In addition to membrane surface protein presentation, a robust understanding of the protein secretions of each oncolytic bacterial species is necessary to assess the anti-cancer capacities that may be added to each species to build on innate abilities. Important details of a species protein secretion system includes knowledge regarding the preferred length or composition (e.g., amino acids, motifs, or hydrophobicity) of exported materials. Gram-negative bacteria primarily accomplish protein translocation through the cytoplasmic membrane corresponding to export into the periplasm, while Gram-positive bacteria secrete proteins directly into the extracellular environment [60]. Gram-positive bacteria have five main protein secretion systems/pathways: Sec (secretory) pathway, Tat (twin arginine translocation) pathway, ABC (ATP binding cassette) transporters, FPE (fimbrilin protein exporter) system, and ESAT-6/WXG100 (early secreted antigen target of six kDa/proteins with a WXG motif of ~100 residues) secretion system [60]. It has been detailed that though Gram-negative bacteria, and likely Gram-positive as well, share many genes corresponding to secretory systems, the secretion pathways present are species, and even sometimes strain, specific [60]. The ABC transporter family of proteins is responsible for many substrates. ABC transporters peptides have only been found in Gram-positive bacteria, making it likely that this system is specific to Gram-positive bacteria [61]. The mechanosensitive ion channel (MscL) permits the release of small proteins (e.g., thioredoxin) during osmotic downshifts [62]. To date, studies regarding the secretome of even the longest studied oncolytic bacterial species, *Clostridia*, seem to indicate a wide range in the amount of secreted proteins [60], indicating that this area of research continues to grow and undergo refinement.

### 1.2. Decades of Efficacy and Safety: The BCG Vaccine

The Bacille Calmette-Guerin (BCG) vaccine against tuberculosis (TB) is commonly administered to infants and young children in developing countries [63]. The vaccine is effective at preventing severe forms of TB, such as TB meningitis, by modulating the immune system response. Specifically, the vaccine drives IFN-γ, IL-12, IL-2, and IL-6 production, effectively inducing a T helper 1 (Th1)-polarized immune response [64]. This same BCG vaccine successfully became an immunomodulatory cancer treatment [65]. Intravesically administered BCG has been the gold standard therapeutic against intermediate and high-risk non-muscle-invasive bladder cancer for almost eighty years [65]. The weakened form of TB bacterium present in the vaccine likely stimulates the production of white blood cells and antibodies capable of tumorigenic bladder cell recognition and destruction through induction of IFN-γ, IL-12, IL-2, and IL-6. When the immune response to the BCG vaccine was characterized, the results indicated live BCG significantly enhanced IFN-γ and IL-12 secretion, expanded CD3^−^CD56^+^ T cells, and increased non-MHC-restricted cytotoxicity against bladder tumor cells compared to unstimulated controls [66]. Given the sustained, successful application of BCG against bladder cancer without toxicity, this therapeutic holds lessons for the translation of other oncolytic bacteria.

## 2. Indirect Oncolytic Bacteria

### 2.1. Klebsiella

#### 2.1.1. *Klebsiella* Basic Microbiology

*Klebsiella* is a Gram-negative, non-sporulating bacteria with facultative anaerobic capacity [67] (Figure 1). This bacterium is non-motile, and tests catalase positive [67]. The average *Klebsiella* cell is 0.3 to 1.0 μm wide by 0.6–6.0 μm in length [67].

#### 2.1.2. *Klebsiella* Genome

The median total length of the *Klebsiella* genome is 5.60 Mbp, with a median GC% of 57.2 [67] (Figure 1).

#### 2.1.3. *Klebsiella* Background and History

*Klebsiella* is commonly found in non-pathogenic states in the human intestines and stool and becomes an opportunistic pathogen in immunocompromised patients [68]. Thus far, the only strain under investigation is *Klebsiella pneumoniae* [1,69]. *Klebsiella pneumoniae* is the most common cause of hospital acquired pneumonia in the United States as it colonizes mucosal surfaces [68].

#### 2.1.4. *Klebsiella* Cell Surface

The oncolytic properties of *Klebsiella* are largely thought to be due to the activation of Toll-like receptors (TLRs) caused by LPS embedded in its cell surface [70] (Figure 2). Activation of the M1 pathway leads to activation of B and T cells in addition to production of cytokines including TNF-α, IL-12, and IFN-γ. The cytokine production promotes inflammation of tumor cells, apoptosis, and inhibition of angiogenesis in the immediate proximity of the tumor [71,72,73]. Capsular polysaccharides (CPS) on the cell surface are also recognized by macrophage TLR-4 receptors and lead to production of TNF-α and subsequent inhibition of angiogenesis, disrupting tumor growth [70,71] (Figure 3). TLR-4 activation on macrophages results in production of IL-1 leading to activation of NF-ĸB and activation of T cells responding to tumor cells [23,25]. The CPS layer within the membrane has also been shown to promote biofilm formation in addition to being a PAMP [1].

Outer membrane proteins, specifically Outer membrane protein A (OmpA), allow for adhesion of the bacterium to host cells and TLR-2 recognition [74] (Figure 3). OmpA is recognized by TLR-2 on dendritic cells and results in inhibition of the M2 macrophage pathway and activation of the M1 pathway [23,74]. OmpA proteins also serve as PAMPs [74], thus activating the M1 pathway, causing activation of B and T cells and production of TNF-α, IL-12, and IFN-γ. These cytokines promote further inflammation, apoptosis, and inhibit angiogenesis [71,72,73]. 

Type 1 and 3 fimbriae expressed on the surface of *Klebsiella* promote adherence to host cells and biofilm formation [75] (Figure 3A). Type 3 fimbriae promotes adherence of the bacterium to the tumor cell forming a biofilm around it thereby disrupting the process of metastasis [75,76] (Figure 3B). The expression of type 3 fimbriae by *Klebsiella* species is indicated to promote biofilm formation and thus increased cell to cell contact [75]. Bacterial biofilms are ubiquitous multicellular aggregates which are usually attached to biotic or abiotic surfaces in which bacteria are embedded in a self-produced extracellular polymeric matrix composed of mainly polysaccharides, proteins, lipids, and extracellular DNA [77]. Biofilm formation is a multistage process started by reversible bacterial adhesion followed by an irreversible attachment with the formation of microcolonies [78]. This may represent an exploitable characteristic for an oncolytic bacteria-mediated oncotherapeutic through the disruption of metastasis by physically coating tumorigenic cells [76] and thus inhibition of further tumor spread. However, biofilms also have the potential to inhibit immune cells from reaching the tumor due to the mixture of polysaccharides, protein, and DNA [79]. Furthermore, the formation of a biofilm has been observed to contribute to antibiotic resistance [76,79]. More studies serving to disentangle the advantages and disadvantages of oncolytic bacterial biofilm formation will be necessary to fully explore the potential role of biofilms in this context.

#### 2.1.5. *Klebsiella* Metabolism and Byproduct Secretion

*Klebsiella pneumoniae* secrete outer membrane vesicles (OMV) containing LPS (Figure 3A). OMVs often detach from the surface, resulting in host immune activation [80,81]. OMVs trigger caspase activation, ultimately leading to the production of IL-1β and IL-8, proinflammatory cytokines that result in inflammation of the tumor cells [80,81] (Figure 3B), which constitutes a particularly promising attribute to exploit for therapeutic delivery. Microcin E492 is a bacteriocin produced by *Klebsiella pneumoniae* observed to induce apoptosis in a range of human cell lines [76,82] (Figure 3A). This bacteriocin is secreted by an ATP-binding cassette (ABC) transporter [83] and functions by inserting into the plasma membrane of host cells to form a pore, thus disrupting cell barriers and ultimately leading to lysis [83]. Microcin E492 has not yet been used as an oncolytic agent as future studies need to be performed to ensure the efficacy and safety of the use of this bacteriocin, but it has potential to be of benefit in future studies.

#### 2.1.6. *Klebsiella* Host–Pathogen Interactions

In pathologic states, *Klebsiella* biofilms infect mucosal surfaces, and can invade cells [84]. Due to its encapsulation, there is increased uptake into the tumor microenvironment and subsequently improved immune system protection [85,86]. As a result of local intracellular invasion, *Klebsiella* can colonize other body surfaces, leading to a severe infection—though this is mostly observed in immunosuppressed and hospitalized patients. Highly invasive, community acquired *Klebsiella pneumoniae* strains typically overproduce capsular polysaccharides, resulting in increased resistance to host clearance [87]. This resistance is not fully understood, but thought to be partially due to hyper-mucoviscosity as a result of increased polysaccharides expressed on the *Klebsiella* surface and secreted into the extracellular environment [87,88]. However, in immunocompetent patients, the presence of LPS and interaction between LPS and TLR-4 signals initiates dendritic cell migration and ultimately T cell activation after antigen presentation [89].

#### 2.1.7. *Klebsiella* Oncolytic Development

An attenuated strain of *Klebsiella pneumoniae* is currently under a Phase II clinical trial for non-small cell lung cancer by Qu Biologics [90], making this oncolytic species a front runner for accomplishing clinical translation. Other studies have used *Klebsiella* in directed, in vivo infections with CBA/J mice, have triggered IL-12 production by macrophages [72,91]. This exposure resulted in Natural Killer (NK) cell and T cell recruitment, as well as IFN-γ production [72]. In vitro studies using human dendritic cells from healthy, human volunteers, demonstrated *K. pneumoniae* fragments containing LPS and outer membrane proteins, including OmpA, were capable of recruiting NK cells, can led to IFN-γ production [72]. NK cells recruitment was driven by pattern recognition receptors specific for OmpA, as well as through CCR-7 expression. CCR-7 responds to CCL-19, a chemokine often found in lymph nodes [72]. *Klebsiella*, if applied as an oncolytic therapeutic, is likely to trigger the immune system reaction in similar ways.

### 2.2. Listeria monocytogenes

#### 2.2.1. *Listeria* Basic Microbiology

*Listeria* is a Gram-positive, non-sporulating bacillus of an average 0.5–4.0 × 0.5–2.0 μm in size (Figure 1) [92]. While *Listeria* can express flagella, this expression and the subsequent motility is temperature dependent and not usually physiologically relevant [92]. Therefore, in the context of bacterial-mediated cancer therapeutics, *Listeria* is considered non-motile. *Listeria* does exhibit catalase activity [92].

#### 2.2.2. *Listeria* Genome

The median reported genome length of *Listeria* is 2.905 Mbp, with a GC content of 38.04% (Figure 1).

#### 2.2.3. *Listeria* Background and History

*Listeria monocytogenes* is the causative agent of listeriosis, a rare foodborne infectious disease with a high and particularly severe incidence in immunocompromised individuals and other risk groups, such as pregnant women, neonates, and the elderly [92]. During a *L. monocytogenes* infection, the bacteria is able to survive in human hosts by invading and multiplying within both phagocytic and non-phagocytic eukaryotic cells [93]. The actin assembly-inducing surface protein (ActA) induces the ability of actin-based motility for *L. monocytogenes* to spread from cell to cell [94], which becomes vitally important for the bacteria to propagate through tissues and evade the host immune system. Listeriolysin O (LLO) is expressed and secreted by *L. monocytogenes* as a cytolysin that disrupts the phagosome and prevents its degradation in the phagolysosome [1] (Figure 4A). Due to its adaptability and unique intracellular life cycle, *L. monocytogenes* has potent potential as a model for oncolytic bacteria.

#### 2.2.4. *Listeria* Cell Surface

The surface proteins internalin A (InlA) and internalin B (InlB) have been identified as the main bacterial factors involved in the invasion of non-phagocytic cells (Figure 4A) [93]. Studies detailing the mechanisms of InlA or InlB-dependent entry have been primarily performed with the human epithelial or trophoblast cell lines [95]. Both internalins bind to the eukaryotic cell membrane receptors E-cadherin and Met [96,97]. InIA interacts with the host E-cadherin, which is present in several human barriers including intestinal, fetoplacental, and the blood–brain barrier [98]. InIA subsequently induces cytoskeletal rearrangement thereby inducing bacterial uptake through receptor-mediated endocytosis (Figure 4B). InIB binds to cellular receptor Met, a tyrosine kinase expressed mainly by cells of epithelial origin, promoting cell invasion and exhibits a much broader range of target cells than InIA (Figure 4B) [97]. Normal hepatocyte growth factor signaling through the Met receptor induces tissue repair that restores tissue structure and prevents fibrosis formation in the liver, kidney, heart, brain, and lung [99]. A fragment of the *L. monocytogenes* InIB was found to have therapeutic potential like that of a full-length hepatocyte growth factor [97], which indicates *L. monocytogenes* has promise for further oncolytic engineering as well as to be harnessed as adjuvant therapy for tissue repair after treatment.

#### 2.2.5. *Listeria* Metabolism, Byproducts and Secretions

Upon *L. monocytogenes* interaction with its host cell receptors, bacteria are engulfed and internalized into a membrane-bound phagosome (Figure 4B). Listeriolysin O (LLO) is expressed and secreted by *L. monocytogenes* as a cytolysin that disrupts the phagosome membrane through generating small pores that uncouple pH and calcium gradients across the membrane (Figure 4B) [1,100]. The optimal pH range of LLO overlaps with that observed in the vacuole, suggesting that LLO may be adapted to function only within the phagosome compartment, protecting infected cells from complete destruction once *L. monocytogenes* is free within the cytosol [101]. This makes *L. monocytogenes* suitable to not only survive but thrive in the acidic environment contained within a solid tumor. Further, an acidic environment plays a key role in the *L. monocytogenes* life cycle as replication and motility can occur in this context without triggering the host immune system [102], intrinsically lowering its virulence. 

LLO pore formations function to manipulate intracellular calcium levels, ultimately leading to a continuous calcium concentration increase until homeostasis is disrupted [103]. LLO at sublytic concentrations induces a broad spectrum of Ca^2+^-dependent cellular responses during infection, indicating *L. monocytogenes* can employ LLO as a kind of remote control to manipulate the intracellular Ca^2+^ level without direct interaction with the host cell [103,104]. Ca^2+^ fluctuation within the cell modulates cellular signaling and gene expression thus providing a potential molecular basis for LLO to disrupt the abundant Ca^2+^-dependent signaling events [103].

Additionally, LLO is necessary for the formation of spacious *Listeria*-containing phagosomes (SLAP), a mechanism that promotes replication within vacuoles [100]. LLO is hypothesized to mechanistically uncouple pH gradients across SLAP membranes, blocking phagosome and auto-phagosome maturation [100]. The replication of *L. monocytogenes* was found to be greatly reduced as compared with those replicating in the cytosol. Initially, in mice with severe combined immunodeficiency (SCID), persistent *L. monocytogenes* infection was thought to imply persistence as chronic infections depended on host immune status [100]. Although LLO expression is required for SLAP formation, SLAPs have been observed in bacterial populations that do not escape from the primary phagosome [100]. Therefore, bacteria within SLAPs may have reduced LLO expression or inefficient LLO activity [100]. LLO in nucleated host cells can also induce cytokine release. In murine cells, the inflammatory cytokines IL-1α, TNF-α, and IL-12—which appeared to be produced mainly by macrophages—were released after *L. monocytogenes* exposure (Figure 4B) [105]. LLO also serves as an antigen recognized by cytotoxic T cells in the context of class I major histocompatibility complex (MHC) molecules [106].

#### 2.2.6. *Listeria* Host–Pathogen Interaction

The extracellular recognition of *L. monocytogenes* PAMPs is performed largely by TLRs—specifically, TLR-2, which is expressed at the highest concentration on myeloid-derived phagocytes (Figure 2) [105]. TLR-5 binds flagellin (Figure 2) [107], which *L. monocytogenes* can possess, but most strains downregulate expression at 37 °C [108]. This temperature sensitive expression could be exploited to gain further biocontainment and thus additional safety for oncolytic therapy. Cytosolic *L. monocytogenes* is thought to be most frequently sensed by NOD-1 and NOD-2 [109]. Sensing of cytosolic *L. monocytogenes* by either of these receptors activates NF-κB and causes increased transcription of proinflammatory chemokines (e.g., CXCL-2) and cytokines (e.g., IL-6, TNF-α) [109].

Intracellular host-sensing of *L. monocytogenes* microbial cell wall components and surface proteins is accomplished through detection by Nod-like receptors (NLRs). Further, because *L. monocytogenes* can gain access to the cytosol of most cell types, any proteins secreted by the bacteria can potentially be processed and presented by the MHC-I machinery [110]. *L. monocytogenes* is therefore an ideal vector for the delivery of tumor neoantigens to be processed and presented, restimulating the suppressed tumor immune microenvironment [111]. Further, this mechanism allows for non-virulent proteins to act as signaling molecules for the immune system, making administration safer—especially when modifications are made to remove virulence factors. The cytotoxic destruction of *L. monocytogenes*-infected cell occurs via one of three distinct pathways l: Fas-FasL-mediated apoptosis [112], TNF-α-dependent apoptosis [113], and granule exocytosis [114]. Lysis of an infected cell likely releases any remaining live intracellular bacteria, which can then be phagocytized by neutrophils or activated macrophages and consequently killed [115].

#### 2.2.7. *Listeria* Oncolytic Development

The unique life cycle of *L. monocytogenes* makes it an ideal candidate to deliver antigens to the MHC I and II pathways, culminating in a tumor-lytic response [116]. Previous studies investigating *L. monocytogenes* as a vector for tumor immunotherapy found antigens were delivered more efficiently to protein processing and presentation machinery than those encoded by other bacterial vectors [117]. Historically, infection with *L. monocytogenes* elicits a strong, long-lasting immunological memory response with protection against future pathogen exposure [118] which may constitute a potential disadvantage as the first exposure may render future treatments non-viable as an anti-cancer therapeutic. When *Listeria* is injected intratumorally, it stimulates the production of pro-inflammatory signals, including reactive oxygen species through the activation of the NADPH-oxidase pathway [119,120]. *L. monocytogenes* has desirable, built in safety features since the flagella of most strains downregulate their expression at 37 °C [108]. The ability to attenuate *Listeria* but retain the potent immune-recruiting capacity is very intriguing and should be considered when considering future developments of bacteria oncolytic therapy.

### 2.3. Mycobacterium

#### 2.3.1. *Mycobacterium* Basic Microbiology

*Mycobacteria* are non-motile, catalase positive, obligate aerobes (Figure 1) [121]. *Mycobacterium* are considered Gram-positive, but in the context of oncolytic activity, they are largely active as intracellular pathogens and thus difficult to Gram-stain [121]. However, *Mycobacteria* stain acid fast due to the presence of mycolic acid in the cell wall and, while variable in length, the average observed size is 0.3–0.5 μm [121]. *Mycobacteria* are not known to form spores.

#### 2.3.2. *Mycobacterium* Genome

The median total genome size of *Mycobacteria* is 5.48 Mbp, with a median GC% 67.1 (Figure 1) [121].

#### 2.3.3. *Mycobacterium* Background and History

There are multiple species of *Mycobacterium* that can be harnessed for their oncolytic properties including the well-known, flagship species *Mycobacterium bovis* (BCG), as well as *Mycobacterium pargordonae* (Pmg), *Mycobacterium indicus pranii* (MIP), and recombinant *Mycobacterium smegmatis* (rSmeg-hMIF-hIL-7) [122]. BCG therapy is used to treat urothelial carcinoma of the bladder, a cancer that was often fatal even with chemotherapy [123]. The application of BCG in this manner began in 1976 and gained its FDA approval in 1990 as it had proven to be effective in decreasing tumor size and preventing recurrence [65,123]. BCG is internalized by local urothelium and inflammatory cells, triggering an inflammatory reaction and cytokine response to the tumor cells [123]. BCG quickly became one of the most successful forms of early immunotherapy and paved the way to harness other bacterial species for their respective oncolytic properties.

#### 2.3.4. *Mycobacterium* Cell Surface

*Mycobacterium* species share several common characteristics that confer oncolytic capacities, including the presence of lipoteichoic acid (LTA) in the cell wall and CpG DNA which, through various mechanisms, lead to activation of the immune system (Figure 2) [25]. In studies with BALB/c mice, CpG motifs within DNA can induce a Th1 response which includes secretion of IL-12 and IFN-γ [124,125]. CpG-DNA is taken up by an immune cell and subsequent recognized and bound by TLR-9 [126], resulting in a signal transduction pathway that ultimately upregulates NF-ĸB and AP-1 transcription factors within macrophages [126]. The use of synthetic oligodeoxynucleotides (ODN) containing CpG motifs (CpG ODN) may be used as an adjuvant to other treatments to further induce production of cytokines and destruction of the tumor microenvironment [124,125,126]. *Mycobacterium* species commonly produce fibronectin attachment protein (FAP), which binds to host fibronectin and plays a critical role in invasion [127,128,129]. Overexpression of fibronectin in the tumor microenvironment often leads to increased angiogenesis and metastasis; this may therefore serve as a target for Mycobacterium expressing FAP and can increase the immune system response due to the ability of the bacteria to infiltrate cells [130,131].

Further, as will likely be of interest for engineering of other oncolytic bacterial strains that are phagocytized prior to accomplishing tumor localization, *Mycobacterium* species produce proline-glutamate or proline-proline-glutamate (PE/PPE) proteins to either evade the host cell or stimulate B or T cells depending on the PE/PPE protein being expressed [132,133]. PE/PPE proteins can be found in the outer membrane to function as nutrient transporters, on the surface of cells, or secreted allowing the *Mycobacterium* cell to interact with the host cell [132,133]. PE/PPE proteins were found to be secreted as a heterodimer and their variable C terminal domains may be what determines the subgroup they belong to and therefore their actions, though the large size has prevented the exact protein structure from being easily identified [132,133]. For example, the PE-PGRS33 protein may assist in macrophagic uptake of *Mycobacterium* and interact with TLR2-promoting dendritic cell activation [132,133]. Additionally, because PE/PPE proteins are expressed on the surface, they serve as epitopes to upregulate the B cell humoral response leading to a systemic response to bacteria [133] and potentially tumorigenic cells. 

In addition to the common characteristics of *Mycobacterium* species, individual species have been studied for their unique oncolytic properties and role in treating various tumors. A brief discussion of each species has been included here:

##### *Mycobacterium pargordonae* (Pmg)

Pmg has been shown to enhance tumor reduction stemming from MC38 cells when administered subcutaneously into nearby lymph nodes in combination with other anti-cancer therapeutics (e.g., cisplatin) in studies on C57Bl/6 female mouse tumor xenograft models [134]. This resulted in enhanced production of IL-12 from dendritic cells and an immune response skewed towards Th1, the proinflammatory state. Pmg also enhances the production TNF-α from CD4, CD8, and NK cells. TNF-α then causes necrosis of tissue, destruction of vasculature, and an antitumor immune response [71].

##### *Mycobacterium indicus pranii* (MIP)

MIP has shown promising results through its use in immunocompetent mouse BALB/c and C57BL/6 models with myelomas (Sp2/0) and thymomas (EL4), respectively [135,136]. In contrast, when used in IFN-γ^−/−^ C57BL/6 mice, a decrease in the production of anti-tumor T cells in correlation with less reduction in tumor size were observed [135,136]. The use of MIP also did not result in reduction in tumor growth in NOD CB17-Prkdc^scid^/NCrCrl (non-obese diabetic severe combined immunodeficient) mice [136]. The reduction in tumor growth was partially due to increased IFN-γ from CD4^+^ T cells and IL-12p70 [136]. Production of IFN-γ has widespread effects within the tumor, causing recruitment of immune cells and induction of apoptosis through FAS ligand and Bcl2 pathways [136,137,138]. IL-12p70 has been known to be secreted early in the immune process to cause inflammation [73]. However, other studies in mouse models have shown that IL-12 also plays a key role in the antitumor immune response through connecting the innate and adaptive response [73], assisting in leukocyte recruitment [139], and inhibiting angiogenesis in tumors [140]. MIP also resulted in an upregulation of CD8^+^ T cells leading to cell mediated killing of target tumor cells [136,141]. A separate study on C57BL/6 IL-10^−/−^ and MyD88^−/−^ mice harvested peritoneal cells and cultured them with heat-killed MIP, live MIP, or BCG [142]. The use of heat-killed MIP resulted in high levels of TNF-α, interleukin-12p40 (IL-12p40), IL-6, and nitric oxide production from the stimulated macrophages [142].

##### *Mycobacterium smegmatis* 

rSmeg-hMIF-hIL-7, a recombinant form of *Mycobacterium smegmatis* is exploited as a delivery vehicle through incorporation of a fusion protein of human macrophage migration inhibitory factor (MIF) and IL-7 into the cell surface. This altered surface protein expression resulted in enhanced humoral and cell-mediated immune responses against the inflammatory cytokine MIF in female mouse models [143]. MIF is commonly overexpressed in many cancers and, when inhibited, anti-human MIF immunoglobulins coreceptors CD74 and CD44 are also downregulated leading to a decrease in cancer cell proliferation and migration [144].

#### 2.3.5. *Mycobacterium* Metabolism and Byproduct Secretion

In contrast to using porin channels to take up nutrients from the surrounding environment, *Mycobacterium* utilize passive diffusion and have adapted to survive in areas of stress and nutrient deprivation [145] such as would be found in a solid tumor microenvironment.

#### 2.3.6. *Mycobacterium* Host–Pathogen Interactions

In pathologic contexts, *Mycobacterium* species can be inhaled from aerosols of water, soil, or surface biofilms. Upon host entry, the bacterium binds to macrophages and proliferates due to its ability to survive intracellularly [146,147]. Subsequently, macrophage function and lymphocyte proliferation are inhibited until the mycobacterium antigens present via MHC molecules to T lymphocytes. This MHC interaction then leads to a systemic immune defense against *Mycobacterium*. Notably, in immunocompromised individuals such as those that may have received immunomodulatory drugs, *Mycobacterium* can spread to the bloodstream and disseminate, causing systemic disease due to inhibition of the production of IFN-γ and downregulated Bcl-2 products in healthy cells, which induces apoptosis of macrophages [148]. Bcl-2 is typically anti-apoptotic, but in its absence, cytochrome c leaks from the mitochondria of host cells and causes caspase activation, degradation of cellular and nuclear proteins, and formation of apoptotic bodies [149].

#### 2.3.7. *Mycobacterium* Oncolytic Development

In contrast, when harnessed for oncolytic capacities, *Mycobacterium* can be administered as a vaccine such as thoroughly demonstrated with the heat killed BCG [1]. In oncolytic directed interactions, the host immune response is re-targeted towards the tumor cells due to a unique intracellular colonization capacity, which recruits immune cells to infiltrate tumor cells as well as tumor microenvironments [134]. Due to the intracellular growth ability of *Mycobacterium* species, they are uniquely able to kill tumor cells through the induction of the inner mitochondrial pathway apoptosis leading to spreading of antigens and immune stimulation against other tumor cells [150,151].

Some *Mycobacterium* species, including *Mycobacterium tuberculosis*, are known to express the ESAT-6 antigen, which when administered as a vaccine intratumorally, demonstrated a reduction in tumor growth thought to be due to activation of dendritic cells and an adaptive immune response against ESAT-6 [152,153]. Using C57BL/6 mice infected with a B16F10 murine melanoma cell line CD8^+^ T lymphocytes were found to be increased in nearby lymph nodes and serum levels of IFN-γ were elevated after anti-ESAT-6 bound to ESAT-6 found on the surface of the infected melanoma cells [154]. The primary use of the ESAT-6 antigen has been on tumors that do not present neoantigens as ESAT-6 is recognized by the immune system upon binding to TLR-2 and major histocompatibility complexes (MHCs) [155].

It is important to note immunosuppressed or immunomodulated patients may not experience the same results when treated with oncolytic bacteria [2]. Use of MIP in immunosuppressed patients did not have the same efficacy in halting tumor growth, indicating that an intact patient immune response is needed for the oncolytic properties to effectively function [135]. This is likely due to a lack of mature dendritic cells, and subsequently a failure to activate T cells, ultimately causing a lack of circulating cytotoxic T cells [135]. A lack of dendritic cells results in the inability to recognize bacterial antigens, and subsequent failure to activated downstream cytokine production pathways—thus the inflammatory immune cells are not produced. Furthermore, in C57BL/6 and IFN-γ^−/−^ mice, the absence of IFN-γ gave rise to a lack of cytokine production [135].

### 2.4. Proteus mirabilis

#### 2.4.1. *Proteus mirabilis* Basic Microbiology

*Proteus* is a Gram-negative, catalase positive bacillus (0.4–0.8 × 1.0–3.0 μm) (Figure 1) [156]. This bacteria is non-sporulating, exhibits flagella-mediated motility, and is a facultative anaerobe [156].

#### 2.4.2. *Proteus mirabilis* Genome

The median total genome size of *Proteus* is 4.06 Mbp, with a median GC content of 38.88% (Figure 1) [121,157].

#### 2.4.3. *Proteus mirabilis* Background and History

*Proteus* is predominantly isolated as a commensal bacterium of the gastrointestinal tract of humans and animals but can cause a variety of infections—most commonly in the urinary tract. The pathogenicity and immune response of *P. mirabilis* can therefore be derived from reports describing the uropathogenic strains [158,159]. *Proteus mirabilis* specifically has been reported to have indirect oncolytic and tumor-suppressive effects [156,160] due to recruitment and reactivation of the host immune system elements. Initial reports indicated *P. mirabilis* was isolated from pus resultant of tumor lysis [161]. *P. mirabilis* RMS-203 (Murata strain) and *P. mirabilis* (Hauser) are currently under development [156,160]. *P. mirabilis* has a sizable collection of virulence factors expressed on the cell surface or secreted during infection [162]. Strategies employed therefore include adherence, toxin production, motility, acquisition of metals, and immune evasion [162]. 

#### 2.4.4. *Proteus mirabilis* Cell Surface

Flagellar motility is a fundamental feature of *P. mirabilis*. As with many bacteria, *P. mirabilis* uses flagella to swim through liquids towards chemical gradients [163]. In liquid cultures, *P. mirabilis* has a short rod shape. However, on rich solid media it differentiates into very long, non-septate polyploid cells with hundreds to thousands of flagella [163]. This gives *Proteus* a signature “swarming” capacity [163]. In addition, antigenic variation through flagellin gene rearrangement enables evasion of host immune response [164]. A *P. mirabilis* mutant lacking cyclic AMP receptor protein (Crp) produced almost no flagella, implying that Crp participates in the surface flagellin protein expression regulation and thus can alter immunogenicity [164]. 

An assorted range of fimbriae are found on the cell surface of *P. mirabilis* that mediate adherence to surfaces like plastic urinary catheters and, more importantly in the context of oncolytic bacteria, host epithelial cells [158]. The most studied fimbriae displayed is the Mannose-resistant *Proteus*-like fimbriae (MR/P) that is capable of eliciting a strong immune response and implicated in both auto-aggregation and biofilm formation [158,159]. The auto-aggregation of *P. mirabilis* is unique in that it forms large clusters that are composed of bacteria, cell debris, and mineral deposits—rendering it well suited to the tumor microenvironment. Cluster formation is dependent on MR/P and urease as in the absence of MR/P the bacteria were found to form small intracellular groups [165]. When looking more specifically at localization of *P. mirabilis* within the bladder, clusters primarily formed on the lumen of the organ wall and indirectly increased the bacterial loads of other bacterial species during colonization. This suggests *Proteus* clusters provide an additional bacterial niche for colonization [159,165,166]. Uroepithelial cell adhesin (UCA) fimbriae facilitate binding to uroepithelial cells and bind to host glycolipids: asialoGM, asialo-GM, lactosyl ceramide, and galectin-3 [167]. The common carbohydrate sequence GalNAcβ1-4Gal commonly present in glycolipids such as those listed is highly targeted by many pathogenic bacteria, making this a possible aspect for future development of bacterial-mediated oncolytic therapy to help drive adherence [167]. *Proteus mirabilis* fimbriae (PMF) was initially studied and found to be important in the localization and colonization of *P. mirabilis* in the bladder but not the kidney [168]. However, it was later identified that it was a necessary fimbriae for both the localization of uropathogenic *P. mirabilis* to the bladder and kidney [169]. Ambient temperature fimbriae (ATF) were observed to be expressed in a culture of Luria broth at 23 °C for 48 h. There was no detection of the fimbriae in the same culture media at 42 °C [170]. ATF does not play a role in colonization or infection, though it likely plays a role regarding *P. mirabilis* survival in the external environment [171]. Harnessing this context-dependent expression and applying this exquisite mechanism to other virulence factors could add an on/off switch for other aspects of oncolytic bacteria, especially if co-administered with exothermic adjuvants.

#### 2.4.5. *Proteus mirabilis* Metabolism, Byproducts, and Secretions

*P. mirabilis* isolates generally possess two important metabolic features that make them ideally suited to growth in human urine: the ability to utilize citrate as a sole carbon source, and the ability to hydrolyze urea to produce an abundant nitrogen source [172]. Urease is a cytoplasmic nickel metalloenzyme that acts by hydrolyzing urea into ammonia and carbon dioxide [172]. The resulting ammonia is the preferred nitrogen source for *P. mirabilis*. In the pathogenicity of a urinary tract infection, the ammonia alkalinizes the urine and can lead to the precipitation of urinary salts to form stones [173]. This may prove useful considering the acidic characteristic of tumor cells and the ability to lower the pH would allow for treatments that were otherwise unsuited to the tumor microenvironment (TME). Further, *Proteus mirabilis* has developed strategies to scavenge the host for micronutrients. There are three distinct mechanisms to harvest iron: proteobactin (Pbt), a non-ribosomal peptide synthesis system (Nrp), and α-keto acids [174,175,176]. 

Hemolysin is a secreted pore-forming toxin that inserts into eukaryotic cell membranes, causing an efflux of sodium ions and cell damage [162]. HpmA is secreted during the mid-exponential or late exponential phase of growth [177] and is the predominant hemolysin secreted by *P. mirabilis*. HpmA was shown to lyse several types of cells including, human bladder epithelial cells, human B-cell lymphoma cells, human monocytes, and African green monkey kidney cells [178,179]. This virulence factor could be used to make hemolysin-mediated pores to allow further entry into the TME. In contrast to hemolysin, *Proteus*-toxic agglutinin (Pta), remains anchored at the bacterial surface and mediates autoagglutination of bacteria and cellular toxicity [180]. Maximum activity of Pta is found at pH 8.5–9.0, with implications regarding the efficiency of activity in the acidic TME [180]. Pta and HpmA have an additive cytotoxicity effect in experimental UTI studies [181] that should be considered in future oncolytic studies using *P. mirabilis*. 

*P. mirabilis* produces ZapA, a metalloprotease with broad specificity. As the enzyme exhibits minimal substrate or amino acid site specificity, it appears that the action of ZapA is limited to antimicrobial peptides as cleavage of IgA, IgG, complement proteins C1q and C3, actin, fibronectin, collagen, laminin, casein, and gelatin have been reported [182,183,184]. ZapA may therefore provide a figurative “weapon of mass destruction”, rather than a highly focused approach, which ultimately may be disadvantageous for oncolytic applications as reducing virulence to protect the host may greatly diminish intrinsic antitumor activity.

Bacterial biofilms are ubiquitous multicellular aggregates which are often attached to surfaces with bacteria embedded in a self-produced extracellular polymeric matrix composed of mainly polysaccharides, proteins, lipids, and extracellular DNA [184]. Biofilm formation is a multistage process started by reversible bacterial adhesion followed by an irreversible attachment with the formation of microcolonies [185]. The ability for *P. mirabilis* to form a biofilm is dependent on a wide range of genes. If successfully exploited, the ability to transfer or implant a biofilm of oncolytic bacteria to a solid tumor may have application as a drug elution patch for targeted drug delivery, thereby limiting systemic effects.

#### 2.4.6. *Proteus mirabilis* Host–Pathogen Interactions

In a UTI mouse model, *P. mirabilis* elevated levels of CXCL1—a potent neutrophil chemoattractant—and IL-10 [186]. *P. mirabilis* formed large clusters in the bladder lumen that drew a massive infiltration of neutrophils [165]. These large clusters and the resulting extracellular matrices ultimately served as protection from the influx of neutrophils [165]. Inflammasomes are protein complexes that form in response to a variety of stimuli and act to induce inflammation [187]. The NLRP3 inflammasome has been specifically linked to intestinal *P. mirabilis*, where a potent IL-1β-mediated pro-inflammatory response was induced. In this setting, NLRP3 activation is dependent on the HpmA toxin [187].

Flagellin is another conserved structure recognized by the innate immune response, specifically TLR-5 (Figure 2) [188]. In the intestine, the Lypd-8 protein, a component of the host defense system that maintains gut homeostasis, prevents flagellated bacteria including *P. mirabilis* from invading colonic epithelium and causing inflammation [188,189,190]. Installation of purified *P. mirabilis* flagellin into the bladder elicits leukocyte infiltration, histological changes in bladder tissue, and elevated CXCL-1, CXCL-10, and IL-6 [188,189,190]. Further, studies have shown Gram-negative bacteria use their LPS core to bind with immune receptors such as dendritic cell-specific intercellular adhesion molecule-e-grabbing non integrin (also known as CD209), as well as human langerin (CD207) expressed by antigen presenting cells. *P. mirabilis* was found to utilize this mechanism to invade CD209-expressing macrophages [191].

A recent study found *Proteus* to be more prevalent in stool and colonic biopsies of patients with diagnosed Crohn’s disease, a disorder due to a dysregulated immune system [192]. Of the isolates, a majority (22/24) of the strains, sequenced were closely related to the uropathogenic strain of *P. mirabilis* [192]. Evidence suggesting *P. mirabilis* is capable of triggering both intestinal and systemic inflammation in murine samples was observed as assessed by colonic myeloperoxidase activity [192]. NOD-like receptor signaling, JAK-STAT signaling, and MAPK signaling pathways were upregulated in response to *P. mirabilis*, alongside NF-ĸB and TNF-α expression as well as IL-18 and IL-1α secretion [192]. However, Ki67-positive signals, a marker of cellular proliferation, were significantly reduced in *P. mirabilis* inoculated mice [192]. Together, these effects culminate in the indirect oncolytic activity of *P. mirabilis* as through activation of the immune system and modulation of cellular proliferation [192].

#### 2.4.7. *Proteus mirabilis* Oncolytic Development

The actual antitumor effects of *P. mirabilis* have been minimally studied. However, a study investigating the mechanistic basis concluded oncolytic capacity is likely exerted by lowering the expression of carbonic anhydrase IX and Hypoxia Inducible Factor-1α (HIF-1α) proteins [160]. Hypoxia is a major challenge within the tumor microenvironment and HIFs mediate the transcription of hundreds of genes that allow cells to adapt to hypoxic environments [193]. This study also observed the expression of NKp46, a cell surface receptor for murine NK cell activation, and CD11c, a marker for dendritic cells—both proteins responsible for priming the immune system—were significantly increased [194]. This elevated signaling may provide confirmation of *P. mirabilis*-induced host cell death through inflammatory cell recruitment [194]. Further, large areas of tumor cell death and inflammatory cell infiltration have been reported after *P. mirabilis* administration [160]. It has been reported that DCs can directly initiate NK cells activation without involving specific cytokines, resulting in innate anti-tumor immune response in the mouse model. This is likely the pathway observed in this study due to the lack of cytokine expression compared to control models [160,195,196].

*Proteus mirabilis* has potential as an oncotherapeutic because it thrives in the harsh environments of the urinary tract. In addition, many aspects from this bacterium have the potential to advance the direction of oncolytic bacteria therapy. For example, the unique ability to form extracellular clusters dependent on urease and the MR/P fimbriae could be utilized as constant “green light” for immune system activation. Further, ATF fimbriae are not expressed at body temperature, and have little understood function in pathophysiology of a *Proteus* infection. Harnessing and applying this molecular signaling with exothermic adjuvant therapy would allow for specific expression patterns of targeted protein secretions or presentation. Hemolysin, the most common secreted toxin of *P. mirabilis*, can be used to create pores in solid tumors to allow for further advancement of current therapeutics into the TME. It is worth noting that one potential disadvantage lies within the non-specific virulence factor ZapA, which likely would have to be properly balanced with anti-tumor effects for safe administration. There is also the potential for *P. mirabilis* to effectively starve the tumor as it has a strong ability to acquire metals.

### 2.5. Streptococcus/Serratia Mix

#### 2.5.1. *Streptococcus/Serratia* Mix Background and History

A mix of *Streptococcus pyogenes* and *Serratia marcescens*, known as Coley’s toxins, demonstrated tumor regression in patients with bone and soft-tissue sarcomas [197,198] and NY-ESO-1 expressing tumors [199]. However, the ratio of bacteria in this mix is unknown leading to mixed views regarding both purpose and efficacy. This unknown administration ratio resulted from a combination of lack of patient follow up, variety in toxin preparations, and changes in methods throughout use of the toxin [197,198]. Consequently, this error set back the entire field of oncolytic bacteria as there was a wide range of toxin responses observed, causing mistrust amongst clinicians and the public. Thus, the failure of Coley’s toxin represents a critical lesson for all oncolytic bacteria progressing toward clinical translation.

Coley’s toxin was originally administered as an injection of heat killed bacteria directly into the primary tumor of patients who were immunocompromised due to use of prior, unsuccessful chemotherapy [197,200]. This simulated an infection within the tumor leading to activation of the immune system and production of cytokines against the tumor without a widespread systemic infection. Further studies of Coley’s toxin led to variable results regarding immune responses against tumors—largely though to be due to frequent high fevers, short treatment duration, and patient immunocompromised status [201]. Despite this, each species expresses different virulence factors that have oncolytic properties of interest for future investigation. The unique combination of these two bacterial species may allow for various immunomodulating effects to target and lyse tumor cells.

#### 2.5.2. *Serratia marcescens*

##### *Serratia marcescens* Basic Microbiology

*Serratia* are Gram-negative, bacillus of 0.5–0.8 μm, facultative anaerobic bacterium that exhibits high motility via swimming and swarming (Figure 1) [202,203]. This species is catalase positive, and non-sporulating [202,203].

##### *Serratia marcescens* Genome

The observed median total genome length for *Serratia* is 5.23 Mbp, with a median GC content of 59.7% (Figure 1) [204].

##### *Serratia marcescens* Cell Surface

LPS in the cell surface activates TLR-4 signaling pathways, which leads to production of inflammatory cytokines that assist in controlling the tumor cell growth (Figure 2) [2,205]. Experiments with B cell lymphocytic leukemia (B-CLL) cells from humans showed that serratamolide, a cyclic peptide within the cell wall, can be cytotoxic to B-CLL cells and induce apoptosis through interfering with survival pathways [206,207,208].

##### *Serratia marcescens* Metabolism and Byproduct Secretion

The virulence factors responsible for oncolytic properties of *Serratia marcescens* are largely prodigiosin, a secondary metabolite [209,210], and the LPS layer found in the outer membrane of the bacterium [211]. Prodigiosin induces p53-dependent apoptosis in multiple types of cancer cells [1,209]. Promoting apoptosis through a p53-dependent pathway is advantageous because a large amount of tumors have p53 inactivating mutations, but it also carries risks because it can affect healthy cells and prevent apoptosis from occurring in cells with DNA damage [212]. Prodigiosin can cause arrest of the cell cycle at the G1/S phase by inhibiting the expression of cyclin [210,213,214].

#### 2.5.3. *Streptococcus pyogenes*

##### *Streptococcus pyogenes* Basic Microbiology

*Streptococcus* are Gram-positive, facultative aerobic cocci of 0.6–1.0 μm in size (Figure 1) [215]. These bacteria are non-motile and non-spore forming, as well as catalase negative [215].

##### *Streptococcus pyogenes* Genome

The *Streptococcus* genome has median total length 1.80 Mbp, with median GC% of 38.4 (Figure 1) [215].

##### *Streptococcus pyogenes* Cell Surface

*Streptococcus pyogenes* expresses LTA, a hyaluronic acid capsule, and M proteins—which function as adhesins allowing microbial attachment to host cells, and subsequently evading an immune response by resisting phagocytosis (Figure 2) [216,217]. LTA is anchored to the cell via its lipid domain and then forms a complex with other surface proteins [216,217]. The hyaluronic acid capsule is the outer layer of the cell and may assist in resistance to phagocytosis, but only when other virulence factors such as the M protein are also present [217]. M protein is a surface antigen responsible for host cell protein interaction allowing for colonization and, for select strains, to adhere to ECM components [218]. The adherence to fibronectin on epithelial cells causes a conformational change, allowing for interaction with integrins and subsequently internalization of the bacteria into the epithelial cell [218]. When looking to develop novel cancer therapeutics, the presence of the M protein ultimately allows for bacterial survival, presentation of the bacterial antigens to immune cells, and production of cytokines against tumor cells [219]. M protein assists in evasion of the immune system through binding complement inhibitory proteins, limiting the complement pathway activation and thereby improving *Streptococcus pyogenes* replication [218]. 

##### *Streptococcus pyogenes* Metabolism and Byproduct Secretion

*Streptococcus pyogenes* synthesizes *Streptococcal pyrogenic* toxins (SPE) [220]: SpeA, SpeB, and SpeC—exotoxins that can bind MHC II and TCR, leading to CD4 lymphocyte activation [221]. Activation of CD4 lymphocytes then leads to cytokine secretion and assists CD8 lymphocytes in mediating the anti-tumor response [221]. *Streptococcus pyogenes* also produces streptokinase, an enzyme that causes plasmin to be released and through a series of mechanisms, results in suppression of vessel formation within the tumor, limiting its growth [222,223]. Streptokinase is also produced by other hemolytic *Streptococci* strains and has been investigated as an adjuvant therapy as cancer cells can activate the clotting cascade and rely on angiogenesis to further growth and spread [224,225].

##### *Streptococcus pyogenes* Host–Pathogen Interactions

In pathologic states, *Streptococcus pyogenes* infects the oropharyngeal mucosa leading to pharyngitis or infects wounds after skin damage [218]. Some proteins produced by *Streptococcus pyogenes* are known as superantigens known to be responsible for heightened immune response that can lead to septic shock [1,218,220]. *Serratia marcescens* is largely isolated clinically from immunocompromised hosts [202,226].

##### *Streptococcus pyogenes* Oncolytic Development

In directed infections, such as oncotherapeutics, a heat killed combination of *Streptococcus pyogenes* and *Serratia marcescens* serves as a mixed bacterial vaccine and can result in stimulation of an immune response [199]. Patients with NY-ESO-1 expressing solid tumors were given subcutaneous injections of the mixed bacterial vaccine (Coley’s toxins) until the maximum dose (547 EU) or maximum desired body temperature (38 °C to 39.5 °C) was reached [199]. At this point, cytokine and IgG levels were measured to determine the level of immune response stimulated [199]. The combination of a facultative anaerobe with an anaerobe in Coley’s toxin may allow for improved tumor cell death. The obligate anaerobe will replicate within the hypoxic tumor environment but will often stay distant from blood vessels [227,228]. The facultative anaerobe will preferentially replicate in the hypoxic environment but can also infiltrate other regions of the tumor including around blood vessels [227,228].

## 3. Direct Oncolysis

### 3.1. Clostridia

*Clostridial* infections have been associated with cancer for centuries—anecdotal observations of tumor regression after a gas gangrenous infection with *Clostridium perfringens* [229] were recorded in 1813. As early as 1927, it was demonstrated that tumor colonization by oncolytic *Clostridia* species could occur without collateral damage to non-tumorigenic cells [230], making them promising novel cancer therapeutics or delivery vehicles. Five species have since been studied for innate anti-cancer or cancer targeting capacities and remain the focus for further development: *Clostridium acetobutylicum*, *Clostridium histolyticium*, *Clostridium novyi*, *Clostridium sporogenes*, and *Clostridium tetani*.

While these bacteria can be a cause of gas gangrene clinically, several techniques (both physical and genetic) have been developed to mitigate, and, in most cases, negate these effects. Most *Clostridia* species are Gram-positive, with a few considered Gram-variable as modulation was consistently observed in vitro [231,232]—though the specific mechanism of Gram-positive to Gram-negative cell membrane conversion remains an open question. Due to flagellation, which allows for migration to and through even the harshest tumor microenvironments that provide a physiological niche for anaerobic bacterial growth, these five *Clostridial* species have particularly promising capacity for systemic (e.g., intravenous) administration. This innate motility is paired with the capacity to sporulate, and for the spore forms to be tolerant of oxygenated environments (e.g. the blood stream, non-tumorigenic tissues), making *Clostridial* species a strong focus for developing further anti-cancer activities. Further, many *Clostridial* species are capable of nitrogen fixation, reducing local nitrogen into ammonia and then biosynthetically incorporation into other metabolically relevant molecules [233].

A brief introduction to each of the five most promising *Clostridia* species has been included for context, with the thorough history of the development of each species detailed elsewhere [229,234]. Here, we discuss the known surface architecture, metabolism, secreted byproducts, and resulting inflammatory stimulation pathways relevant for anti-cancer capacities or development of such capacities as well as highlight promising preliminary data for each oncolytic strain. Extensive discussion of *Clostridium novyi*-NT, including Figure 5, is featured as a representative of this species due to its recent advancement to Phase III clinical trials [235].

#### 3.1.1. *Clostridium acetobutylicum*

##### *Clostridium acetobutylicum* Basic Microbiology

*C. acetobutylicum* are 0.5–0.9 × 1.5–6 μm rod shaped bacteria with motility due to peritrichous flagella, and Gram-variable [236]. These bacteria are sporulating, generating oval and subterminal spores [236]. They are obligate anaerobes, and can only survive hours in an oxygenated environment before sporulation is forced, with no catalase activity [236].

##### *Clostridium acetobutylicum* Genome

*C. acetobutylicum* has 4 Mbp genomic chromosomes with 11 ribosomal operons [237], a ~200 kb plasmid with pSOL1 (solventogenesis genes) [237]. The loss of pSOL1 leads to degeneration [237], which has important implications for exogenous plasmid transformation in this species. The average genomic GC content it 30.9% [237].

##### *Clostridium acetobutylicum* Background and History

*C. acetobutylicum* has several advantageous characteristics that suggest potential as an oncolytic bacterium for further development. First, it is largely considered nontoxigenic and nonpathogenic in both the spore and vegetative forms, lending an inherent level of safety for use in a clinical setting [237]. Next, a wide range of methods have been optimized and published in detail [238], allowing for ease of modification, and in addition, the entire genome has been sequenced [236,239], along with a putative proteome and secretome [60], allowing for the development of further genetic engineering techniques. Perhaps most uniquely, *C. acetobutylicum* has a characteristic extracellular cellulosome [60] that could be exploited to contain exogenous enzymes, pro-drugs, or therapeutics. *C. acetobutylicum*, with a robust body of literature in biofuel applications, is considered a model organism among *Clostridia* [240] with a high proteomic percent identity to the other oncolytic *Clostridia*. Importantly, while *C. acetobutylicum* has been found in the human colon, it is thought to be entirely benign, and not typically detected in human microbiomes [241]. Due to its benign nature, the only threat for this strain is if pathogenic, plasmid-encoded genes are acquired from toxigenic *Clostridium* species. While this is extremely rare, and preventative clinical strategies can be easily implemented, there have been documented incidences [242] that cannot be ignored.

*C. acetobutylicum* has been widely applied in the field of biofuels due to its characteristic solventogenesis where in mono-, di-, and even some polysaccharides (e.g., starch) are converted efficiently into acids and solvents, such as butanol [238]. Specifically, *C. acetobutylicum* has demonstrated an advantageous capacity for production and secretion of individual cellulosome components [238]. This body of literature can be used to extrapolate *C. acetobutylicum* characteristics within the context of oncolysis as the same characteristics make it an excellent candidate for development as consolidated bioprocessing units, or when a microorganism strain conducts all major synthesis steps to product to a commercial product [238] such as biofuel or pharmaceuticals. In large part because of the substantial investment into biofuel development, several genetic modification methods have been optimized and published in detail [238], including systems with replicative, constitutive, and inducible promoters as well as whole gene knock-in/out systems.

##### *Clostridium acetobutylicum* Cell and Spore Surface

Using several available databases [60], many membrane-associated proteins were identified via bioinformatic analysis of the *C. acetobutylicum* genome [236,239,243]. Briefly, permeases, PTS components, enzymes (e.g., sortase and peptidases), signal transduction systems, protein secretion system components, and adhesions were identified, as well as many proteins that have not yet been annotated. At least three holin-like proteins were found, all without detectable N-terminal signal peptides [60]. Holins are involved in the secretion and activation of enzymes with muralytic activities that appear to hydrolyze the cell wall prior to cell lysis with a primary role in the transport of specific lytic enzymes (largely those lacking N-terminal signal sequences) [60,244] that likely play a role in oncolytic activity.

Further, ten ESAT-6-like proteins [245] have been reported in *C. acetobutylicum* [60,236], all with conserved WXG motifs but no putative function—though the WXG motif implies a role in bacterial virulence [60]. ESAT-6 family genes encode small proteins known to be potent T cell antigens fundamentally important to bacterial virulence secreted extracellularly [246]. Many *Clostridial* species have demonstrated the secretion of hydrolytic enzymes and toxins, such as collagenase [247], which could be used to anchor engineered proteins to confer added oncolytic capacities.

Surprisingly, but perhaps advantageously for development of anti-cancer capacities, the expression of a non-cellulolytic cellulosome of >665 kDa has been implied [60]. Additionally, some evidence of both pili biogenesis genes and transporter pathways were indicated. While the role of fimbriae is reported to have a virulence role in pathogenic *Clostridia*, their role has yet to be elucidated in *C. acetobutylicum* [60]. Similarly, while flagellar assembly has been thoroughly detailed in Gram-negative bacteria, it is just beginning to be characterized in detail for Gram-positive bacteria such as *Clostridia* [248,249]. It is worth noting that when *C. acetobutylicum* is exposed to butanol, the lipid concentration of the membrane is altered, increasing acyl residue content and subsequently shifting the membrane to be more rigid to compensate for the disruptive effects of butanol [250]. Hypothetically, similar environmental contexts would be encountered when administered as a cancer therapeutic.

##### *Clostridium acetobutylicum* Metabolism, Byproducts, and Secretions

Optimal growth of *C. acetobutylicum* occurs at 37 °C, and requires biotin and 4-aminobenzoate as growth factors [251]. *C. acetobutylicum* can use several carbohydrates found within the tumor microenvironment to generate energy, and has demonstrated that molecular attractants include butyric acid and sugar [252]. Unsurprisingly, *C. acetobutylicum* motility is deterred by acetone, butanol, and ethanol as this would generate movement away from metabolic byproducts towards nutrient sources [252].

Notably, different growth phases of *C. acetobutylicum* produce different byproducts [233] which has implications for the potential secretion of oncolytic therapeutics. This metabolic transition is an adaptive process that results from the effects of acid toxicity that would presumably also result from the acidic tumor microenvironment. Further, as the motility of *C. acetobutylicum* has been demonstrated to be a chemotactic response [252], common characteristics of the tumor microenvironment [250] are likely to influence migration capacities as well. Spore formation in *C. acetobutylicum*, in contrast to the canonical endospore formation of *Bacillus subtilis*—which is initiated when nutrients become limited—is initiated due to aerobic conditions [253]. Germination can then be induced by the reverse environmental conditions [253].

Bioinformatic analysis of the *C. acetobutylicum* genome [60,236,243] identified proteins necessary for the FPE and ESAT-6/WXG100, but not those for the Tat protein secretion pathway. This is perhaps unsurprising since the Tat pathway was absent in *C. tetani* [254] as well, and thus its absence is hypothesized to be a genus feature [60]. Further, the major proteins known to be ABC transporters were not identified in *C. acetobutylicum*, though a single ABS transporter was identified as a member of the lipoprotein translocase (LTP) family [60], perhaps due to the Gram-variable nature of this particular species.

Several proteins known to be responsible for peptide export belonging to the Pep1E (peptide-1 exporter), Pep2E (peptide-2 exporter), Pep4E (peptide-4 exporter), Pep5E (3-component peptide-5 exporter), and bETE (b-Exotoxin I Exporter) families were identified [60]. While no homologue for MscL (mechanosensitive ion channel) was identified by this study, sequences corresponding to a Sec system, flagella export apparatus (FEA), fimbral protein expore (FPE), tight adherence (Tad) system, several holins, and an ESAT-6/WXG100 secretion system were found [60]. This study seems to imply that a number of virulence factors, including phospholipase C, virulence factor MviN, hemolysins, and adhesins, are secreted by these systems [60], which is particularly surprising since *C. acetobutylicum* is largely considered non-pathogenic. Finally, a number of proteins with cell adhesion domains were observed, as was the presence of a Tad system [60]. Together, these components are likely to play critical roles during colonization as well as pathogenesis [60].

As in biofuel production, oncolytic *C. acetobutylicum* mainly secretes butanol, though also notably ethanol, acetate, and hydrogen [60]. Butanol can be toxic if produced in high enough quantities, though the necessary quantity would be extremely high [255], as it causes cellular membrane permeability to protons and subsequently intracellular ions increase, proton dynamics dissipate, and energy conduction fails [60]. The intracellular pH control is in turn suppressed, which leads to penetration of intracellular macromolecules (e.g., RNA phospholipids and proteins—making it possible that this capacity could be harnessed to deliver gene therapy components. It is important to note that the toxicity of butanol is not selective to cell type, and many studies have encountered self-toxicity. Intriguingly, associations have been indicated between motility and sporulation with solvent production (e.g., non-motile mutants do not produce solvent) [249,252,256] that could prove advantageous for further development as well as to address presumed safety issues, perhaps even in a way that is translatable to other pathogenic *Clostridial* species.

Biofilm capacity has been indicated for *C. acetobutylicum* under a particular growth context [77]; however, these contexts are unlikely to occur within tumors. Intriguingly, a *C. acetobutylicum* biofilm produces extracellular polymeric substances rich in heteropolysaccharides (e.g., glucose, mannose, aminoglucose) and cytoplasmic proteins [77], as well as many non-classically secreted proteins acting as non-canonical adhesins (GroEL, rubrerythrin). Continuous mode operation of a *C. acetobutylicum* biofilm significantly increased productivity with the biofilm capable of acting as an enzyme reservoir for external processing of substrates [77]. Most proteins localized to this biofilm, such as molecular chaperons and stress proteins, are not typically related to physiological processes with many thought to be intracellular (e.g., GAPDH, TPI, pyruvate-ferredoxin oxidoreductase, electron transfer flavoprotein, and alcohol dehydrogenase) [77]. The capacity for *C. acetobutylicum* to conduct non-classical secretion [77], when paired with the expanding methodology for genetic engineering could prove particularly advantageous for anti-cancer capacity development. Further, when continuously grown in a biofilm, *C. acetobutylicum* indicated a decreased ability to initiate sporulation [77]. This lack of sporulation could be exploited to overcome some of the purported safety hurdles regarding off-target localization and environmental impact on the journey toward clinical translation.

##### *Clostridium acetobutylicum* Host–Pathogen Interactions

Beyond canonical Gram-variable host–pathogen stimuli (e.g., PAMPs, LTA, LPS) previously discussed, *C. acetobutylicum* is considered non-pathogenic and non-toxigenic in both the vegetative and spore form [60].

##### *Clostridium acetobutylicum* Oncolytic Development

While *C. acetobutylicum* does not have innate direct oncolytic activity, the robust methodology of genetic engineering from the biofuel literature has given rise to strains capable of producing enzymes to activate pro-drugs with extremely specific localization to the anaerobic environment within solid tumors [257]. *C. acetobutylicum* was engineered by the introduction of recombinant plasmids to exogenously produce mouse tumor necrosis factor alpha (mTNF-α) at significant levels while retaining bioactivity—notably in both cell lysates and secretions [258]. Thus, this study elegantly plotted a course forward for the engineering of *Clostridial* species with additional oncolytic capacities. 

Further, *C. acetobutylicum* strains have been modified to induce oncolytic activity through stimulation and modification of the immune system. The strain DSM792 has been engineered to produce rat interleukin-2 (rIL-2) with therapeutic efficacy [259]. IL-2, a pluripotent cytokine known to non-specifically enhance immune response (e.g., natural killer and lymphokine-activated killer cell activation) and major histocompatibility complex (MHC) restricted T cell responses that result in clearance of neoplastic cells, has been successfully used to treat metastatic melanoma and renal cell cancer, although at high systemic doses [259]. High systemic doses of IL-2 demonstrate several severe and serious side effects [259], yet it is still used clinically in specific cases. However, these side effects were avoided through direct delivery to the tumor site by exploiting the physiological niche contained within to attract *C. acetobutylicum* [259].

#### 3.1.2. *Hathewaya histolytica*, Formerly Known as *Clostridium histolyticum*

##### *Hathewaya histolytica* Basic Microbiology

*H. histolytic* forms Gram-positive 3–5 μm × 0.5–0.7 μm rods, clumped in pairs or short chains with motility due to rich flagellation [260]. These bacteria are sporulating, with asacchrolytic, proteolytic endospore formation [260]. *H. histolytic* is catalase positive [260].

##### *Hathewaya histolytica* Genome

*H. histolytic* is reported to have a 2.7 Mbp genome and is currently undergoing characterization [261].

##### *Hathewaya histolytica* Background and History

*Clostridium histolyticum* has been recently (2016) reclassified as *Hathewaya histolytica*. However, since this reclassification was relatively recent, most of the published literature referenced within this review used the former *Clostridial* designation. Further, many modern publications, particularly regarding commercial collagenases isolated from *Hathewaya histolytica* (Xiaflex), continue to use the name *Clostridium histolyticum*. Given the nature of this review, we have chosen to keep *Hathewaya histolytica* within the *Clostridial* section due to the high level of similarity in both genome and proteome. These characteristics have profound impact on oncolytic characteristics and how to confer additional anti-cancer capacities. To maintain transparency from the historical literature and in case of future classification alterations, we will use the classification originally published within this section’s sources.

The first peer-reviewed, documented evidence of *Clostridial* spores anti-cancer capacity was demonstrated in 1947 when *C. histolyticum* was injected into murine sarcomas [260]. Subsequently, lifespans were extended by as much as twenty days beyond untreated control mice [260]. Unfortunately, none of the tumors were completely destroyed and after a penicillin treatment to eradicate remaining *C. histolyticum*, large sarcomas developed from the remaining tumor margins [260]. Intriguingly, in the same study, isolated secreted toxins from *C. histolyticum* culture (presumably exotoxins) resulted in marked tumor regression when injected locally and repeatedly [260]. However, the required therapeutic dosage proved to be near lethal [260]—perhaps due to the thin line between exploiting inflammatory immune reactions for tumor mitigation versus over stimulation of immune cells initiating septic cascades. This study importantly determined that for this *Clostridia* species, tumor regression was due to the production and secretion of proteolytic enzymes [260].

*Hathewaya histolytica* is considered a toxigenic species due to a potent secretion of exotoxins with both proteolytic and necrotizing capacities capable of instigating severe localized necrosis [262,263]. Yet, intriguingly, reports of human infections are extremely rare [264]. It should be noted that the infrequency of documented infections could be in part due to inherent difficulty in clinically isolating anaerobic bacteria, though *H. histolytica* is not thought to be a strict anaerobe. *H. histolytica* can cause gas gangrene with symptoms of pain, fever, muscle necrosis, and massive edema that can ultimately results in sepsis, multiorgan failure, and death if not treated [262]. Escalation of this infection can progress rapidly, becoming life-threatening within hours of exposure [262].

*H. histolytica*, in most cases, can be treated with clindamycin, penicillin, metronidazole, imipenem, as well as a cocktail of antimicrobials commonly used clinically to target mixed species necrotizing infections [262]. It is of utmost importance to maintain awareness of the ‘normal’ clinical risks associated with oncolytic bacteria, as missteps such as those resulting in outbreak or conferring antibiotic resistance would have widespread impact, potentially not only stagnating the field, but contributing to decades of clinical and public resistance to the general idea of live biologic therapeutics.

##### *Hathewaya histolytica* Cell and Spore Surface

A single electron microscopy study has been conducted on *C. histolyticum* sporulation and germination processes, indicating no architectural or procedural differences from that observed for *Clostridial* species [265]. This study observed an outer spore coat composed of three layers, with at least architectural similarity to other *Clostridial* spores. Beyond this single, eighty-year-old reference, we were unable to find any other publications regarding the *C. histolyticum* cell or spore surface proteins or putative proteomes. *C. histolyticum* does not appear to have had a full genome sequence published or made widely available, though the colH gene sequence has been reported [266] in early efforts to isolate *C. histolyticum* collagenase. *Hathewaya histolytica* was reported to a single entry in the NIH GenBank under BioProject number PRJEB18332 in 2019, alongside data that this genome has a 34% GC content level, and the genome is thought to be around 2.8 Mbp [267]. A UniProt database entry regarding the *H. histolytica* exists [261], though with an ‘Unreviewed’ designation from the database, and largely without peer-reviewed analysis as well as low annotation scores.

##### *Hathewaya histolytica* Metabolism, Byproducts, and Secretions

*C. histolyticum* secretes a mixture of collagenases, proteolytic enzymes, and exotoxins with a unique efficiency for converting tissue proteins into amino acids and peptides [263]. Unusually, at least in relation to other *Clostridial* species, *C. histolyticum* is not known to metabolically produce isoacids such as isobutyric or isocaproic acids, but rather seems to have a preference for acetic acid [263]. *C. histolyticum* has demonstrated an ability to digest casein, gelatin, hemoglobin, albumin, collagen, and elastin [263], through secretion of at least five toxins: alpha, beta, gamma, delta, epsilon. Further, these enzymes are capable of not only digesting tissue, but soft parts of bone as well [263], indicating potential efficacy against tumors found in bone should adequate safety be achieved. 

Alpha-Toxin

Strains of *C. histolyticum* that produce high levels of alpha-toxin have been recorded to cause death in animal models within 24hr of inoculation [263]. Similar to *C. novyi*, when alpha-toxin is absent no toxemia is observed, however, tissue destruction still occurs and is thought to be the result of secreted collagenolytic and proteolytic toxins [263]. Alpha-toxin is considered necrotizing but not hemolytic, and can be neutralized by antisera [263]. It is possible that this neutralization capacity may be of value for future oncolytic development as well as translating to other species with similar secretomes. Further, not all isolated strains of *C. histolyticum* produce alpha-toxin in culture, and the functional structure seems to be rather unstable and readily inactivated by proteolytic enzymes [263].

Beta-Toxin

Beta-toxin, in contrast to alpha-toxin, has been identified as a family of seven collagenases [263]. Generally, collagenase is a zinc metalloprotease responsible for cleavage of native triple helix collagen as well as gelatin into small fragments. These seven collagenases have both Class I and Class II collagenase activities hypothesized to give rise to synergistic digestion [263]. Beta-toxin is capable of inducing hemorrhage and edema even when purified away from the other exotoxins secreted by *C. histolyticum* [263]. It is worth noting that collagen remodeling often accompanies solid tumor formation, and the impact of this remodeling on cancer progression and metastasis is currently being studied. Preclinical studies are beginning to observe the behavior of *C. histolyticum* collagenase (CHC, commercially known as Xiaflex) within the context of breast cancer, though published efforts remain extremely preliminary [268]. When directly injected, CHC can degrade collagen structures [269].

Gamma-, Delta-, and Epsilon Toxins

As a thiol-activated proteinase, gamma-toxin can digest hide powder, azocoll, gelatin, and casein but demonstrates no activity against collagen [263]. Like alpha-toxin, delta-toxin has proteolytic activity, but with specificity for elastase [263]. Further, this toxin is readily inactivated by reducing agents, but not chelating agents [263]. Epsilon-toxin is an oxygen-liable hemolysin with intriguing similarity to hemolysins produced by other *Clostridium* species.

Other proteases have also been reported to be secreted by *C. histolyticum*, such as clostripain, lethal toxin, and high-potassium-sensitive toxin—all considered to be cytotoxic [270]. Clostripain is a heterodimeric cysteine endopeptidase with strict specificity for Arg-Xaa peptidyl bonds [271]. When this gene was expressed in *E. coli*, the enzyme retained capacity to hydrolyze clostripain substrates [271]. Overall, it seems the enzyme activity observed largely depends on which strain of *C. histolyticum* is present as well as the environmental context of the infection [270]. Interestingly, combination of *C. histolyticum* secreted toxins with common protease inhibitors may have increased cytotoxicity towards in vitro HeLa cells rather than inhibiting or blocking action [270].

##### *Hathewaya histolytica* Host–Pathogen Interactions

Beyond the direct implications of the *C. histolyticum* toxic secretions, details of host–pathogen interactions have not been reported. This may in part be due to safety issues regarding experimentation with this species, as well as the short-time frame that has been observed between *C. histolyticum* exposure and death [263].

##### *Hathewaya histolytica* Oncolytic Development

Very little modification, genetic or otherwise, has been reported to either *C. histolyticum* or *H. histolytica*. There is likely, and understandably, hesitation to experiment with this species due to the safety implications of its potent exotoxin secretions.

#### 3.1.3. *Clostridium novyi*

##### *Clostridium novyi* Basic Microbiology

*C. novyi* is considered Gram-variable, being Gram-positive in young cultures but Gram-negative as the culture ages (Figure 1) [231]. These bacteria are rod-shaped (7–10 mm), catalase negative, bacteria with the capacity to form spore of 2–5 μm in size. *C. novyi* are ultra-strict anaerobes, with motility due to peritrichous flagella on the vegetative cells [231] as well as spores [272] (Figure 5).

##### *Clostridium novyi* Genome

The *C. novyi* genome is 2.55 Mbp (Figure 1), with the toxin responsible for sepsis, α-toxin, contained on separate phage-DNA plasmid [273], which has important implications for safety [231] and exogenous plasmid transformation in this species. The average genomic GC content is reported to be 28.9% [273].

##### *Clostridium novyi* Background and History

In 2001, twenty-six strains of anaerobic bacteria were probed for their capacity to grow and spread within the solid tumor microenvironment specifically to exploit these bacteria for their diagnostic and therapeutic capacities [231]. *C. novyi*, along with *C. sordellii*, was not only able to survive the harsh tumor microenvironment but to spread throughout the tumor, including poorly vascularized regions when intratumorally injected into a subcutaneous B16 mouse model. Further, upon intravenous injection, *C. novyi* accomplished tumor-specific localization, germination, and subsequent colonization within 16hrs of inoculation [231]. Due to the characteristic exquisite sensitivity to oxygenated environments, *C. novyi* colonization of a tumor is halted at the oxygenated margins [274], which provides an extra layer of control and subsequent safety [274]. However, 16–18 hrs post-treatment, all of the mice died from suspected lethal toxins released by *C. novyi* germinating in the tumor microenvironment [231]—which is common in anaerobic bacterial infections. Thus, to mitigate systemic toxicity, the toxin responsible was removed to achieve safety alongside efficacy [231]. While *C. sordellii* had a similar tumor growth capacity, *C. sordellii* has two homologous toxin genes, but the *C. novyi* lethal toxin, α-toxin, is contained within a phage episome [231]. Thus, *C. novyi* α-toxin can be removed via a simple heat treatment [231] to permeabilize the cell membrane, and subsequent work both validated and added efficacy to the originally reported method [275]. This new modified strain retained the capacity to localize to tumors upon IV administration and germinated selectively within the microenvironment, but did so in a non-toxic manner, earning the designation *Clostridium novyi*-Non Toxic (NT) [231].

A subsequent study in immunocompetent mouse models found more than thirty percent of mice treated with *C. novyi*-NT were considered cured, despite the oxygenated tumor margins remaining post-germination [274]. In contrast to what was reported for combinatorial treatment with current chemotherapies in an immunodeficient model (nude mice), animals treated with *C. novyi*-NT spores exhibited complete tumor regression independently [274]. This result was supported by similar results in a rabbit VX2 tumor model, wherein a cure rate of 30.4% was observed [274].

##### *Clostridium novyi* Cell and Spore Surface

Relatively few studies have been conducted regarding the chemical composition of any spore coat, but a study was conducted on *C. novyi*-NT spores using atomic force microscopy (AFM) and ultrathin transmission electron microscopy [272]. Briefly, *C. novyi*-NT spores consist of (from the outside inward) a sacculus, an amorphous shell with intertwined honeycomb layers, often a honeycomb layer attached to the coat, approximately three to six coat layers, undercoat, cortex, germ cell wall, and the spore core [272]. Occasionally, via AFM, a spore outer membrane was detected beneath the undercoat with an inner spore membrane also sometimes found below the germ cell wall [272]. Typically, the amorphous shell formed by retained mother cell cytoplasm remains present after sporulation [272]—though likely varies with the isolation methodology employed. Notably, evidence of an exosporium layer covering the amorphous shell was not detected [272], though, again, this could be due to processing prior to imaging.

While this study reports a complex, multilayer architecture with several significant landmarks [272] for reference, it was not capable of determining composition beyond broad generalization. Bacterial spore coat assembly is largely thought to be a self-assembly process similar to crystallization and thereby influenced by conditions such as salt concentration, pH, and environmental impurities rather than merely biochemical pathways leading to spore coat protein production [272]. In turn, the minute alterations in the coat could influence spore resilience, persistence, and germination capacity among other characteristics [272]. The proteins forming *C. novyi*-NT spore coat layers demonstrated characteristics that make them unlikely to be globular and are thought instead to be stretched peptides protruding from the layers [272]. When this structure is observed elsewhere (e.g., paraffin, fat crystals) it results in a biomaterial with relatively strong, hydrophobic interaction between long neighboring units but relatively weak interactions between the layers [272].

Further, as this type of packing involves hydrophobic interactions, a high proportion of the peptide components likely contain hydrophobic amino acids [272]. This hydrophobicity may be responsible for the difficulty in dissolving spore coat proteins [272] and should be noted for those looking to modify the spore surface. Layers seem to be ‘pinned’ together by screw dislocations, which serve to make the spore coat a cohesive entity woven together rather than separate layers merely deposited on top of each other [272]. In situ spore coat degradation is likely facilitated by germination activated lytic enzymes, which can be found readily in the genome [273]. Further detailed knowledge of spore coat composition would undoubtedly help determine future genetic modifications of *C. novyi*-NT.

##### *Clostridium novyi* Metabolism, Byproducts, and Secretions

As *C. novyi* is anaerobic, it does not use oxygen to break down nutrients but rather produces fatty acids and other organic compounds. Genomic analysis indicates that *C. novyi* is not highly related to any of the five *Clostridial* species (*C. acetobutylicum* [236], *C. botulinum* [276], *C. difficile* [277], *C. perfringens* [278], and *C. tetani* [254]) that have previously undergone genomic analysis. Of the 1255 coding sequences found in the *C. novyi*-NT genome, 551 are unique to *C. novyi*-NT, including 139 with no known prokaryotic homologs [279]. Despite this, 153 of the proteins that were putatively identified are predicted to be at the cell-surface or secreted, with many identified as potentially cytolytic with an ability to degrade lipids or proteins [279]. This includes a homolog with a high percent identity to a *C. perfringens* phospholipase C (NT01CX0979), known to possess activities such as hemolysis and arachidonic acid cascade activation, which subsequently initiate a series of inflammatory responses in the host [273]. While lipases are usually not considered cytolytic, one *C. novyi* lipase (NT01CX2047) has demonstrated an additional ability to alter lipid bilayers, change membrane permeability and thereby potentiate cytotoxicity (Figure 5B) [280]. It is highly likely that the *C. novyi* phospholipase C contributes to both direct and indirect tumor destruction through induction of inflammatory cascades, which is supported by high expression both in vitro *C. novyi*-NT cultures as well as *C. novyi*-NT isolated from tumors and thus in an infectious state (Figure 5B) [273].

Spores have been generally thought to be metabolically dormant, including the absence of mRNA. Surprisingly, *C. novyi*-NT spores have been shown to contain mRNA [273]. However, rRNA was not only found within *C. novyi*-NT spores rigorously purified on a Percoll gradient, but the rRNA species of spores were found to be different from that of the vegetative cells [273]. Further, a number of transcripts in *C. novyi*-NT spores were found to be significantly different from that of vegetative cells—though the role of these transcripts remains unclear [273]. Some of the transcripts are enriched for enzymes known to mitigate reactive oxygen species, as would be required for successful germination. Interestingly, the most abundant mRNA species found in *C. novyi*-NT spores have no known homologs [273].

Finally, *C. novyi* insertion sequence elements (ISE) seem to represent a novel form of transposition termed “crucitrons” [273]. In contrast to canonical transposition, *C. novyi* ISE involves an endonuclease activity that recognizes and cleaves a stem-loop structure formed at palindromic sequences [273]. Subsequent *C. novyi* ISEs then insert specifically into these cruciform structures [273]. This adaptation represents a particularly advantageous form of gene insertion as it does not disrupt the transcriptional termination site, which are instead perfectly duplicated during transpositions, avoiding disruption of an essential gene [273]. *C. novyi*-NT has a relatively small genome when compared to its *Clostridia* cousins [273], making this a particular challenge for synthetic genetic modification. Crucitrons therefor may represent a rather unique mechanism for transposition that poses no threat to the host, and isn’t subsequently influenced by host transcription [273].

##### *Clostridium novyi* Host–Pathogen Interactions

After *C. novyi*-NT colonization of subcutaneous CT26 tumors in immunocompetent BALB/c mice, serum cytokines indicated the presence of IL-6, MIP-2, G-CSF, TIMP-1, and KC [274]—all polypeptides associated with neutrophil chemotaxis and activation. Further experimentation indicated an accumulation of inflammatory cells at the margins of tumors colonized by *C. novyi*-NT largely composed of neutrophils [274,281]. Seventy-two hours after treatment, the ring of inflammation included monocytes and lymphocytes alongside neutrophils [274]. Further, to establish whether the inflammatory response to *C. novyi*-NT had given rise to an immune response against the tumor, mice were re-challenged with CT26 cells in the opposite flank [274]. Surprisingly, 80% of the mice were resistant and rejected tumor implantation [274]. This unique property may have application for tumor microenvironments that are considered to be particularly immunologically “cold”. Previous attempts at immunotherapy in relatively large tumors has been challenging [282]. These results were supported by further experimentation in a VX2 rabbit tumor model, where in animals cured of a liver tumor by *C. novyi*-NT treatment were re-challenged with an intramuscular tumorigenic cell injection that failed in all subjects [274].

To further probe *C. novyi*-NT immune stimulation, tumor bearing mice were treated with either surgical excision to reduce the tumor burden without large-scale immune stimulation, or *C. novyi*-NT spores [274]. When mice were considered “cured” of the original tumor (1–3 months post-treatment), mice were rechallenged with a subcutaneous CT26 tumor [274]. Of the surgically cured mice, 83% developed tumors, but a mere 20% of the *C. novyi*-NT treated mice developed tumors, supporting the hypothesis that *C. novyi*-NT infection has an adjuvant-like effect to stimulate an immune-mediated antitumor response [283]. This hypothesis garnered further support when the results were repeated in a second subcutaneous tumor model using RENCA tumorigenic cells where 33% of the mice were cured, and all cured mice demonstrated complete resistance to a second tumor challenge [274]. Intriguingly, when *C. novyi*-NT treated mice cured of RENCA tumors were rechallenged simultaneously with both a RENCA cells in the right flank and CT26 cells in the left flank, the RENCA tumors failed to establish but the CT26 tumors developed at the same rate as in a naïve mouse [274]. Adoptive transfer of CD8+ cells from mice previously cured of their tumors via *C. novyi*-NT treatment prevented tumor establishment for all naïve mice subjected to a CT26 tumor challenge without being exposed to *C. novyi*-NT themselves [274]. However, adoptive transfer of CD4+ cells from previously *C. novyi*-NT cured animals did not prevent tumor establishment [274]. Further, this systemic immunity appears to be long-lasting as up to 11-months after initial treatment rabbits cured by *C. novyi*-NT treatment retained the capacity to reject subsequent tumor challenges [274].

While these pre-clinical data are encouraging, it is difficult to predict if the immunostimulatory results will recapitulate in human studies as human tumors display an array of inherent immunogenicity, protective capacities, and resistance mechanisms. However, given that *C. novyi*-NT has demonstrated germination in mice, rabbits, canines, and preliminary human studies as well as in human xenografts within nude mice (notably deficient in T cell-mediated immunity) [274], this attribute of *C. novyi*-NT mediated oncotherapeutics is particularly valuable to address the climbing cancer incidence and death rates.

##### *Clostridium novyi* Oncolytic Development

Early studies probed the efficacy of *C. novyi*-NT when combined with conventional chemo [284] and radiotherapeutic agents to treat the tumor from both its core and the tumor margins. Xenograft tumors of several tumorigenic cells were conducted in nude mice [231,284]. Flavone acetic acid and microtubule-binding agents (*i.e.*, vinblastine, vincristine, colchicine, combretastatin A-4, D10) were administered in combination with *C. novyi*-NT, with D10 producing the most pronounced effects [231,284]. Indeed, the anti-tumor effects of combinatorial treatment, or COBALT (combination bacteriolytic therapy), proved to be quite dramatic, with targeted hemorrhagic necrosis evident within twenty-four hours and shrinkage occurring within 2–4 weeks [231,284]. Many necrotic masses were fully mitigated, leaving mice tumor-free with limited deaths due to toxicity reported [231,274]. These combinatorial results were rarely observed in this model when *C. novyi*-NT spores were administered alone; however, it is worth noting that this tumor model used immunodeficient nude mice that lack adaptive immunity. Later studies, as discussed previously, indicated that the adaptive immune response(s) stimulated by *C. novyi*-NT targeted infection are critical for mitigating tumorigenic cells within the oxygenated margins as well as conferring life-long renewed tumor surveillance [274].

Recent studies have pioneered using modern genetic engineering (e.g., CRISPR/Cas mediated genetic engineering) to modify the spore surface of *C. novyi*-NT to overcome premature clearance [275]. It is worth noting that many studies are underway to elegantly harness the exquisite intrinsic sensitivity of *C. novyi*-NT in novel tumor imaging strategies as well [285,286], though these studies are beyond the scope of this review.

#### 3.1.4. *Clostridium sporogenes*

##### *Clostridium sporogenes* Basic Microbiology

*C. sporogenes* is a Gram-positive, catalase positive, rod shaped bacteria of 0.3–1.4 × 1.3–16.0 μm [287]. This anaerobic bacteria is capable of both sporulation and flagellar motility [287].

##### *Clostridium sporogenes* Genome

The typical *C. sporogenes* genome is 4.1 Mbp with a 27.8% GC content as well as an additional 16.3 kb plasmid [287], which has important implications for exogenous plasmid introduction. Intriguingly 16S rRNA displays 99.7% sequence similarity to *C. botulinum* strain A, with additional 2016 orthologs [287].

##### *Clostridium sporogenes* Background and History

*C. sporogenes* was first isolated from soil and classified as the non-pathogenic *Clostridium butyricum* M55, then renamed *Clostridium oncolyticum*, and is now known as *Clostridium sporogenes*. Due to the non-pathogenicity of this strain, early studies hypothesized that this strain would retain the advantageously specific tumor localization of its cousins, but without the high levels of toxicity [288]. Indeed, as predicted, *C. sporogenes* localized and germinated specifically in a solid Ehrlich tumor model [288]. This localization and subsequent colonization generated extensive but specific lysis—at the time termed “liquefaction” [288]. Though not all the mice survived the extensive oncolysis, this study is an important landmark because tumor destruction occurred without damage to local normal cells, which would have been especially impressive given the alternative treatments at the time [288]. Unfortunately, tumors eventually regrew from the remaining tumor margins [288].

Unsurprisingly given the large body of literature with detailed studies, *C. sporogenes* was the first *Clostridia* strain to be reported to undergo genetic engineering [289]. The introduction of a gene encoding a *E. coli* bacteriocin with oncostatic capacity, Colicin E3, was reported, though with heavily nuanced and complicated results [289]. Ultimately, it was discovered that *C. sporogenes* did not have favorable characteristics for genetic modification methods of the time [289]. Like most *Clostridia* species, transformation and thus genetic editing is difficult but possible, and it is critical a well-suited, customized system be employed specific to the context of the desired outcome.

##### *Clostridium sporogenes* Cell and Spore Surface

Typically, *C. sporogenes* presents in long chain-like clusters of cells with a cell wall composed of a thick peptidoglycan layer displaying protrusions of other components, such as LTA and extracellular proteins, giving it a more rigid structure [287]. While there is significant similarity to *C. botulinum*, *C. sporogenes* does not produce botulinum neurotoxins [287] and thus is considered relatively safe. However, *C. sporogenes* has been reported to cause hemorrhage in rabbits specifically, and the highly proteolytic nature may act as an adjuvant for opportunistic pathogens [232]. It is currently hypothesized that the toxins responsible for hemorrhage in rabbits cause either an increase in the permeability of the blood capillary wall or otherwise directly act up endothelial cells to cause destruction [232]. Given that the hemorrhage was localized exclusively to the outermost organ surfaces, the toxin does not easily penetrate serosal membranes nor into blood circulation [232]. Intriguingly, this hemorrhagic capacity is not demonstrated in humans or other laboratory animals [232].

##### *Clostridium sporogenes* Metabolism, Byproducts, and Secretions

As obligate anaerobes, like the strains discussed above, *C. sporogenes* conducts Stickland fermentation, or the fermentation of amino acids, avoiding the use of hydrogen ions as electron acceptors [290]. Glucose fermentation is possible, but considered to be a secondary source of nutrition [290]. Notably, with applications for oncolytic development, *C. sporogenes* is an acidophile with a preferred pH environment of 6.0–7.6 [290]. Some strains have indicated an ability to produce bacteriocin-like secretions capable of inhibiting other *C. sporogenes* strains [291], and possibly other *Clostridia* as well. When found in the human gut microbiome, *C. sporogenes* converts tryptophan into the potent antioxidant indole-3-propionic acid (IPA) [292,293]. IPA is currently under investigation as a novel therapeutic for neurodegenerative disease [294] as well as cancer [295].

Some studies seem to have indicated that even heat-treated *C. sporogenes* can retain its natural cytotoxicity [296] while adding further safety attributes as the bacteria are ‘dead’. When monolayer cultures of the colorectal cancer cell lines CT26 and HCT16 were exposed to heat-inactivated *C. sporogenes*, significant reduction in the cancer cell proliferation and viability rates were observed [296]. Further, administration of media that had been previously cultured with *C. sporogenes* and therefore presumably contains exotoxins but not the bacteria itself (“conditioned” media) correlated the results of the heat-treated bacterial study [296]. However, monolayer cell culture has limited translatability to in vivo models, and thus further testing explored heat-treated *C. sporogenes* and conditioned media against 3D tumor spheroids [296]. After just 72 h spheroids exposed to inactivated bacteria or conditioned media were observed to be smaller than control, untreated spheroids—including an overall regression of size as well as altered morphology indicative of tumor cell death [296]. While extracellular matrix breakdown was observed paired with morphological integrity loss of spheroid cultures [296], neither the proteins responsible nor the mechanism of this activity has been elucidated. Nonetheless, this study has very interesting implications for added levels of safety for the development of *C. sporogenes* as well as other *Clostridial* species that secrete exotoxins.

##### *Clostridium sporogenes* Host–Pathogen Interactions

The hemorrhagic capacity of *C. sporogenes* has been studied in mice, rats, guinea-pigs, rabbits, and in many monolayer and three-dimensional cell culture models by purifying the toxin responsible [232]. Only rabbits were observed to have a macroscopic accumulation of blood and fluid in the peritoneum and, though to a lesser extent, the pleural cavities [232]. Notably, even much higher dose-administration of the purified hemorrhagic toxin failed to cause observable pathological changes. However, details regarding the mechanism of these effects or the protein responsible have not yet been studied.

##### *Clostridium sporogenes* Oncolytic Development

The results of the first *C. sporogenes* studies were overshadowed by the researchers’ decision to demonstrate lack of pathogenicity by self-injection, luckily without harmful effects [288]. With this “evidence” of safety, five neoplastic patients were admitted to the first documented clinical trial of intravenously administered oncolytic bacteria in 1964. Three of these patients indicated oncolysis in the largest of their tumors, but not in metastases or surrounding tissues [288]. In 1978, a second clinical trial was conducted, with administration occurring via intracarotid injection for glioblastoma patients [297]. Within a week, most patients were observed to have complete oncolysis with the conversion of the tumors into abscesses, which were then operated on to mitigate harmful side effects [297]. Side effects of low-grade fevers only occasionally required supportive care of fluid administration and antibiotic treatment [297]. Ultimately, though this study demonstrated both safety and efficacy, an overall lack of clinical benefit was declared due to regrowth from the well-vascularized, tumorigenic rim of cells that remained after treatment [297]. The incomplete mitigation of tumors alongside off-target inflammation leading to sepsis became a common theme leading up to the 1980s, when researchers began to attempt combination therapy, or combining the administration of *Clostridial* spores with other therapeutics to create varying degrees of synergistic effects [234]. However, promising results from these initial studies were further complicated by a purportedly significant difference in response depending on which animal model was used [289]—though this might be explained by the immune state of the model rather than species, especially as complexities of the immune system were just beginning to be elucidated.

Several *C. sporogenes* strains have been genetically modified to incorporate therapeutic proteins with a specific focus on enzymes that sensitize the tumor to specific chemotherapeutic agents with thorough reviews found elsewhere in the literature [229,257,298,299], as has their progress toward clinical translation [57]. This allows for both specific targeting to tumor cells exclusively, as well as potent ‘bystander’ effects as nearby tumor cells are also exposed to the therapeutic drug [300]. Collectively these strategies have been called directed-enzyme-prodrug therapy, or DEPT [300]. Antibody or gene incorporation into this strategy has then been referred to as ADEPT and GDEPT, respectively [300], though this terminology has not been widely implemented in the field. Indeed, a lack of standardized language and therefore search terms to refer to oncolytic bacteria therapeutics may be hindering progress toward clinical translation.

#### 3.1.5. *Clostridium tetani*

##### *Clostridium tetani* Basic Microbiology

*C. tetani* are rod shaped (0.5 μm × 2.5 μm), Gram-positive bacteria with motility due to flagellation [254]. These strict anaerobes grows best 33–37 °C, have the capacity to sporulate and are catalase negative [254].

##### *Clostridium tetani* Genome

The median *C. tetani* genome is 2.8 Mbp with a 75 kbp pE88 plasmid that encodes tetX and its direct transcriptional regulator, TetR [254]. The median GC content 24.5% [254]. Intriguingly, 82% of the predicted ORFs observed are transcribed in the same direction as DNA replication [254] which is atypical for common pathogens. Further, very few mobile elements (16 transposases) were detected [254]. Lack of GC content fluctuation indicated that lateral gene transfer events accomplishing gene acquisition have not occurred in recent evolutionary events, which means that the *C. tetani* genome is more stable than typically observed for enteropathogenic genomes [254].

##### *Clostridium tetani* Background and History

The first peer-reviewed studies probing oncolytic capacity of *C. tetani* is often considered the biggest setback for the oncolytic bacterial field. When spores were administered intravenously in a mouse tumor model, all treated animals died of tetanus within 48hrs [301]. Intriguingly, despite the widespread lethality, microscopic examination of tissues in subjects indicated germination of *C. tetani* occurred exclusively in cancerous tissues, with the vegetative, antigenic form remaining within tumors [301].

##### *Clostridium tetani* Cell and Spore Surface

In contrast to *C. perfringens*, the *C. tetani* genome analysis indicated an array of surface layer proteins. Several homologues (19) to the *C. difficile* adhesin Cwp66 were observed, all with multiple copies of the putative cell wall-binding domain PF04122 [254]. At least two proteins with multiple leucine-rich repeat domains similar to Internalin A, a *L. monocytogenes* protein known to mediate host epithelial cell binding, were detected [254]. Interestingly, at least 11 surface layer putative proteins were identified that lack homologues in any other sequenced *Clostridial* genome, but indicate percent identity with characterized surface protein domains such as bacterial Ig-like domains [254]. Further, the sequences of these proteins include a leader peptide thought to indicate extracellular exportation [254]. Surface layer protein genes were noted to be clustered near each other in the genome, which may be of interest for future genomic modification to add safety [254].

##### *Clostridium tetani* Metabolism, Byproducts, and Secretions

A general, thorough *C. tetani* metabolic network can be found in another review [302], but certain attributes are worth highlighting within the context of oncolytic therapeutic development. When the genome of *C. tetani* was compared to *C. perfingens* and *C. acetobutylicum*, it was discovered that *C. tetani* can uniquely rely on an extensive sodium ion bioenergetic pathway [254]. Twenty-seven peptidases were discovered via genetic analysis, many constituting putative zinc-metalloproteases [254]. Seven of these peptidases are known to facilitate lipid degradation, fifteen for ethanoloamine utilization, and twenty-one for amino acid decomposition [254]. Combined with a lack of the typical genes responsible for metabolizing sugar, the presence of these genes culminates in an intriguing and unique ability among *Clostridial* species to use amino acids as an energy source [302]. While *C. tetani* can process many amino acids, it lacks biosynthetic pathways for at least phenylalanine, histidine, isoleucine, lysine, leucine, methionine, tryptophan, and valine, causing amino acid auxotrophy [302]. This attribute is not uncommon for pathogenic bacteria, particularly those with relatively small genomes as their host typically readily provides these amino acids [302]. Furthering this unique nature, *C. tetani* does not have the genes for the F_0_F_1_-type ATPase present in *C. acetobutylicum* and *C. perfringens*, but rather a type V ATPase is used to synthesize ATP during fermentative metabolism [254]. The sodium ion membrane motive force created by this mechanism is important to note during efforts toward oncolytic development because at least 6 multidrug resistance exporters are driven by sodium ion intrusion [254].

A Sec-dependent secretory mechanism signal peptide was detected to be included in 419 putative *C. tetani* proteins [254]. At least 101 of these observed proteins indicate involvement in transport, though 160 strongly hydrophobic proteins have no known function [254]. N-terminal signal peptides are predicted for virulence and surface proteins, peptidases, sporulation-specific proteins and sensory proteins responsible for chemotaxis among other processes [254]. It is therefore likely that the well-characterized Sec-system is the major mechanism of protein translocation in *C. tetani* [254]. Evidence of a signal-recognition particle-like pathway known to be responsible for mammalian protein translocation was also observed [254]. This system likely plays a role essential for both protein translocation and protein insertion into membranes [254], both important processes to understand when developing additional oncolytic capacities.

##### *Clostridium tetani* Host–Pathogen Interactions

*C. tetani* is well known to be the cause of tetanus, which manifests as spastic paralysis [254]. Tetanus is caused by the tetanus toxin released by *C. tetani* with a very low lethal dose of about 1 ng/kg [254]. *C. tetani* is commonly found in soil, and therefore often in animal intestinal tracts as well [254]. The nearly ubiquitous nature of this pathogen necessitated the development of a vaccine that constitutes part of the common immunization schedule for most developed countries [254]. Regardless, more than four hundred thousand cases of tetanus were recorded in 2003, with most occurring in neonatal patients too young for vaccination [254]. In tetanus disease, the tetanus toxin (TeTx) blocks the release of neurotransmitters from presynaptic membranes of inhibitory neurons of the spinal cord and brainstem, thus catalyzing the cleavage of the synaptic vesicle protein, synaptobrevin [254,303]. TeTx thereby elicits the canonical muscle contractions, including “lockjaw” that characterizes tetanus disease [254,303].

The same plasmid, pE88, that encodes the genes for Tetx also encodes a collagenase virulence factor, CoIT [254]. This enzyme plays a critical role in pathogenesis as a direct agent of host tissue integrity destruction [254]. pE88 further encodes seven regulatory proteins known to govern the tetanus toxin formation with some notable similarity to TxeR toxin regulation in *C. difficile* [254]. In addition to the virulence factors encoded in pE88, genes encoding tetanolysin O, hemolysin, and fibronectin-binding proteins were found [254]. Full details of all the virulence genes discovered in a genetic study of the *C. tetani* genome have been elegantly summarized elsewhere in the literature [254].

It is worth noting that *C. tetani* is industrially grown to produce the common vaccine against tetanus disease [302,304]. However, when media variations occurred, particularly in the casein digest components, batch-to-batch variability in the toxin titers has been demonstrated [304]. A loss of sporulation capacity was also observed [304]. This variability has been largely overcome with stringent, chemically defined media recipes [302]. Both the hurdle and method to overcome this challenge towards clinical translation have been previously reviewed [56] with applications to *C. tetani* development as well as other oncolytic bacteria.

##### *Clostridium tetani* Oncolytic Development

A single study of the oncolytic capacity of *C. tetani* occurred in 1955 [301]. In this study, intravenous administration of *C. tetani* spores in CSH/He mouse models of breast cancer (CSHBA) fibrosarcoma (HE 8971) and hepatoma (98/15) [301]. Two female BALB/c mice bearing spontaneous mammary tumors were also inoculated. This treatment resulted in the death of all subjects within 48 h, regardless of tumor size, type, or dosage [301]. However, intriguingly, non-tumor control mice survived without displaying symptoms of tetanus [301]. It is therefore likely that spore germination followed by the production of the tetanus toxin occurred exclusively in cancer tissue [301]. This hypothesis was supported by microscopic examination of tumor and normal tissue sections [301]. The high and quick lethality observed in this study has led to stagnation in the development of this particular *Clostridial* species for its oncolytic effects.

### 3.2. Salmonella

#### 3.2.1. *Salmonella* Basic Microbiology

*Salmonella enterica* serotype *typhimurium* is a Gram-negative, flagellated facultative anaerobic bacilli that is characterized by serological positive H, O, and Vi antigens and whose only known reservoir is the human body (Figure 1) [305,306,307].

#### 3.2.2. *Salmonella* Genome

The typical *S. typhimurium* genome is 5 Mbp encoding around 4000 genes, more than 200 of which are functionally inactive (Figure 1) [308].

#### 3.2.3. *Salmonella* Background and History

Several *Salmonella* strains have demonstrated potential for use as a bacteria-mediated cancer therapy, but perhaps the most promising is *S. typhimurium* [309]. *S. typhimurium* has many advantages over other oncolytic bacteria including: high tumor specificity, native cytotoxicity, ability to readily modify the bacteria’s gene profile, deep TME penetration, and favorable safety profiles with attenuation [310]. *Salmonella* is known to colonize solid tumors where they can proliferate and cause direct oncolysis as well as stimulate an immune response in a difficult to penetrate environment. *S. typhimurium* in particular has been engineered for use in cancer targeted therapies [311]. Despite the characteristically low oxygen saturation of the TME, this harsh environment provides a unique niche for *Salmonella*, abundant with nutrients that are released from apoptotic cells [309,312]. Further this environment has immunosuppressive conditions with chemokine signaling that promotes growth and colonization in tumor tissues for specific salmonella strains [313]. *S. typhimurium* infections have also demonstrated an inhibitory effect on VEGF expression of, which is known to stimulate the formation of blood vessels [150,310,313], thus providing vasculature to deliver canonical oncotherapeutics. Noticeable benefits have been shown via the use of engineered/attenuated strains for the treatment of solid tumors, some of which have been granted FDA approval for early-phase clinical trials [311].

VNP20009, a strain of *S. Typhimurium*, is the most studied strain capable of targeting solid tumors in part due to its characteristically excellent safety profile [314]. Safety mechanisms of this strain include attenuation of virulence via a deletion in the *pur1* gene, decreased potential septic shock via a deletion in *msbB* gene and antibiotic susceptibility above that demonstrated by the unattenuated VNP20009 strain [314]. These genes are necessary for Lipid A and adenine synthesis [314]. VNP20009 is the only *Salmonella* strain to be evaluated in a phase 1 clinical trial for treatment of nonresponsive metastatic melanoma or renal cell carcinoma in humans [315]. However, the *S. typhimurium* VNP20009 was unable to colonize tumors at a substantial enough level to elicit antitumor effects potentially due to over attenuation [316].

#### 3.2.4. *Salmonella* Cell Surface

Targeted *Salmonella* infections have demonstrated upregulation of connexin 43 expression, a molecule responsible for promotion of the gap junction formation between tumor cells and adjacent dendritic cells. Such gap junctions allow for free passage of preprocessed tumor proteins to dendritic cells for presentation via MHC class I favoring CD8^+^ anti-tumor lymphocytes. Moreover, gap junctions also are required for differentiation of B and T Cells, antibody section via B cells, dendritic cell activation and T-regulator cell activity [317]. Thus, connexin 43 upregulation aids in activation of B and T cell lymphocytes [318].

#### 3.2.5. *Salmonella* Metabolism, Byproducts, and Secretions

*S. typhimurium* A1-R was developed specifically to grow in xenograft tumors by treating *S. typhimurium* with nitroguanidine (NTG). The addition of NTG induces amino-acid-auxotrophic attenuated mutations for leucine and arginine, giving rise to selective growth in environments rich in these amino acids, which includes tumor xenografts [319,320]. Auxotrophs are organisms that have lost the ability to synthesize certain substances required for growth due to the presence of mutations [321]. Exposure to the A1R strain resulted in high tumor targeting efficacy and limited proliferation in normal tissue—largely thought to be due to these mutations [319,322,323]. A1-R has demonstrated inhibitory effects on a wide variety of patient-derived xenograft (PDX) murine cancer models including: prostate, pancreatic, breast, lung, ovarian, cervical, stomach, melanoma, gliomas, and sarcomas [316,319,322,323,324,325,326,327,328,329,330,331,332,333,334,335,336,337,338].

The attenuated *S. typhimurium* ΔppGpp is a strain that has been modified to removed ppGpp (SL) synthesis demonstrated selective proliferation and colonization in solid tumors while showing a significant increase in lethality as compared to its wild type strain [339,340,341,342]. When this strain was further engineered to express Cytolysin A (ClyA), it decreased pancreatic tumor size in athymic Nude mice with both subcutaneous xenografts as well as orthotopic tumors (Figure 6A) [343]. ClyA is a native bacterial toxin produced by *S. typhimurium* that importantly exhibits oncolytic activity between tumor tissue and stromal cells via a pore forming mechanism [341,344]. This pore forming toxin permeates the neutrophilic barrier as well as the proliferating areas of the tumor, ultimately reducing the tumor growth [310]. In the presence of L-arabinose, the ClyA protein was specifically found in tumor tissues harboring the SL^lux/ClyA^, confirming that expression of ClyA was specific to pancreatic cancer tissues. SL^lux/Cly^ expression of ClyA is only noted during the presence of L-arabinose, and thus is absent without this addition [343].

#### 3.2.6. *Salmonella* Host–Pathogen Interactions

The typical host–pathogen response to a *S. typhimurium* infection begins with TLRs recognition of PAMPs typical of Gram-negative bacteria as has been discussed previously (Figure 2). The antitumor activity exerted by *S. typhimurium* is mainly induced by TLR4 activation while the role of TLR5 role remains auxiliary (Figure 6B) [345,346]. *Salmonella* infections elicit a large cytokine and chemokine storm that can ultimately lead to an immune cell influx into the TME, but can also result in severe toxicity [347]. It has been observed that ΔppGpp *S. Typhimurium* can attract immune cells using specific markers, such as CD45^+^ for leukocytes, CD11c^+^ MHCII^+^ for dendritic cells, CD11b^+^ F4/80^+^ or CD68^+^ for macrophages, CD3^+^ CD8^+^ for CD8^+^ T cells, CD11b^+^ Gr1^+^ or Ly-6G/LY6C^+^ for neutrophils and B220/CD45R^+^ MHCII^+^ for B cell (Figure 6B) [339]. Analogous to viruses, *Salmonella* has demonstrated anti-tumor capacities by intrinsic oncolytic properties such as the SopE2, SseB and SseD exotoxin secretion (Figure 6A) [348]. *Salmonella* accumulation in solid tumors further deprives the tumor cells of extracellular nutrients, thus promoting tumor cell apoptosis [309,312]. Upregulation of TNF-α is often observed, ultimately leading to hemorrhage and eventually the inhibition of tumor angiogenesis [349]. Tumoral vessel destruction is augmented by high tumor vascularity due to *Salmonella* exotoxin activity [326]. Moreover, *Salmonella* has also demonstrated inhibition of tumor angiogenesis through suppressing levels of VEGF and HIF-1α in the AKT/mTOR pathway (Figure 6B) [350].

Immunomodulatory activity is also demonstrated by *S. typhimurium*, with adaptive immune response also playing a strong role [310]. Antitumor activity is driven by two main pathways: (1) upregulation of immunostimulatory factors such as IL-1β, IFN-γ, and (2) inhibition of immunosuppressive factors by arginase-1, IL-4, TGF-β, and VEGF (Figure 6B) [339,340,346,351]. Migration of innate immune cells including macrophages, dendritic cells, and neutrophils are enhanced by expression of pro-inflammatory cytokines such as TNF-α, IL-1β, and IL-18 to result in overall tumor regression [310]. Release of these cytokines amplifies both the local and systemic immune response [352,353,354,355]. TNF-α upregulation due to *S. typhimurium* tumor colonization increases permeability of tumoral blood vessels, and can cause hemorrhage, enhancing the influx of immune cells and increasing the anti-tumor effects [349,356]. Resulting tumor cell damage then causes ATP release, which in turn activates inflammasomes, specifically NLRP3, which elicits further increase the circulating level of pro-inflammatory cytokines [342]. Two mechanisms have demonstrated inflammasome activation in targeted *S. typhimurium* infection: (1) direct activation via TLR-4 (LPS) and/or (2) activation via ATP being released by the damaged tumor cells (Figure 6B) [342]. It is important to note these mechanisms are not mutually exclusive, and have been observed to act in tandem [310]. Activation is therefore ultimately achieved through ATP binding of the P2 × 7 receptor on inflammasome macrophages [342].

*S. typhimurium* has a demonstrated capacity for intracellular survival and replication—particularly in macrophages, dendritic cells, phagocytes, and neutrophils [357,358]. Exposure to *S. typhimurium* induces host metabolism changes, some of which promote *S. typhimurium* proliferation. First, glucose levels decrease when glycolysis is upregulated glycolysis to increase bacterial uptake of carbon sources such as 2 and 3-phosphoglycerate and phosphoenolpyruvate, thus accumulating glycolytic intermediates [358]. This is observed alongside a decrease in serine levels, the effects of which are mediated by SopE2, a bacterial protein that leads to the accumulation of carbon sources needed for replication and proliferation [358]. Secondly, an increase in pyruvate and lactate levels leads to stimulation of SPI-2, which is known to encode virulence factors for *S. typhimurium* [358]. Such virulence factors from SPI-2 include, SseB and SseD which are filament and pore forming components of the secretion apparatus for *S. typhimurium* (Figure 6A) [359]. Migration via the immune system allows for systemic translocation, most commonly to the liver, spleen, and bone marrow [360]. Researchers have noted after infection, *S. typhimurium* initially colonized the spleen and liver, but three days post IV inoculation, colonization and proliferation began to preferentially invade the tumor environment in the presence of L-arabinose [341,345].

#### 3.2.7. *Salmonella* Oncolytic Development

Tumorigenic cells have been characterized to accomplish metastasis more readily when in the G_0_/G_1_ phase as DNA is tightly packed away and therefore unavailable to the mechanistic damage caused by most chemo- and radiotherapies [332]. Thus, tumorigenic cells in the G_0_/G_1_ phase could be protected from cytotoxicity, which in part leads to developing resistance to these therapeutics [332]. Previous research has shown that 90% of the total cancer cells in the center and 80% of established solid tumors are in the G_0_/G_1_ phase as cells are either not dividing (G_0_), or are preparing to divide (G_1_) [332]. Intriguingly, *S. typhimurium* A1-R demonstrated an ability to alter cell cycle stages from G_0_/G_1_ into S/G_2_/M, generating sensitivity to chemotherapeutic agents due to the renewed availability of DNA [332,361]. This suggests A1-R is able to maintain antitumor activity without eliciting toxicity to other host tissue due to its high affinity for cell in the G_0_/G_1_ phase [322].

Observations in xenografts of the human pancreatic ductal adenocarcinoma cell line, AsPC-1, in BALB/c athymic nu^−^/nu^−^ (nude) mice indicated colonization rates of tumors were significantly higher than in the liver and spleen [343]. Tumoral stroma plays a large role in metastasis, invasion, and prognosis of solid tumors—but especially pancreatic ductal adenocarcinoma. Therefore, the ability to penetrate and destroy this stroma represents an advantage over current therapeutics [310]. Further, Matrix metalloproteinase 9 (MMP-9) induced in tumor microenvironments contribute to tumor progression and metastases. *S. typhimurium* can inhibit the expression of MMP-9 in TME’s via the downregulation of the AKT/mTOR pathway preventing epithelium to mesenchymal transition [362].

## 4. Prophylactic Bacteria

In addition to oncolytic bacteria that have demonstrated direct or indirect activity against tumors as described above, a growing body of literature reports another group of bacteria with anti-cancer capacities that does not fit into either of these categories. These bacteria include species commonly employed as probiotics for gastrointestinal health and other microbiome alterations. *Bifidobacterium*, *Caulobacter*, and *Lactobacillus* have displayed an ability to mitigate the inflammation and signaling contexts thought precede solid tumor initiation, potentially “derailing” tumor formation. In this context, these bacterial species could be considered prophylactic, and in much the same ways as described previously, could be engineered. These prophylactic, or cancer-preventing species, represent another promising avenue for implementing bacteria as therapeutics and thus have been included in this review.

### 4.1. Bifidobacterium

#### 4.1.1. *Bifidobacterium* Basic Microbiology

*Bifidobacterium* are Gram-positive, rod shaped anaerobes (Figure 7A) [363]. This species is non-spore forming, non-motile, and catalase-negative [363].

#### 4.1.2. *Bifidobacterium* Genome

The median total length reported for the *Bifidobacterium* genome is 1.98 Mbp, with median GC content of 67.9% (Figure 7A) [364].

#### 4.1.3. *Bifidobacterium* Background and History

*Bifidobacterium* species are naturally present in the human intestinal microbiota [363,365,366,367], and commonly found in probiotic and dairy products, like milk and yogurt [363]. Like most oncolytic bacteria, *Bifidobacterium* has a demonstrated, innate ability for tumor localization and thrives within the hypoxic and immunosuppressive microenvironment of a solid tumor. *Bifidobacterium* has a unique ability that differs from other oncolytic bacterial species in that it activates the immune system while having low to almost no toxicity, and is therefore generally regarded as “safe” (GRAS) by the FDA [368,369]. For example, *B. infantis* had no observable cytotoxicity when cultured in vitro with erythrocytic, healthy hepatocytic (LO2), or lung (BEAS-2B) cell lines [368]. Further, when injected into BALB/c models, no significant effect on biochemical parameters or functions of the heart, liver, spleen, lung, or kidney were detected [368].

*Bifidobacterium* has demonstrated what could be considered a probiotic, protective role against pathogens and gastrointestinal diseases by maintaining the integrity of the intestinal barrier and permeability of tight junctions between intestinal epithelial cells [366]. In vitro studies exposing human Caco-2 monolayer cells to LPS resulted in a decrease in the tight junction proteins occluden, claudin-3, and ZO-1, and therefore increased disorganized tight junction protein structure and damage [366]. However, treatment of the damaged cells with *Bifidobacterium* restored protein expression to within the range of healthy controls, decreasing the permeability of the tight junctions [366]. This restorative activity was confirmed in neonatal necrotizing enterocolitis mouse models, where *Bifidobacterium* repeated its attenuation of the damage from necrotizing enterocolitis [366]. Further, analysis of human gut microbiota revealed that those positive for *Helicobacter pylori* infection, a known cause of gastric and duodenal cancer [367], had significantly lower abundance of *B. longum*, *B. adolescentis*, and *B. bifidum* [367]. These results imply a greater presence of *Bifidobacterium* in the gut can be protective against a known carcinogenic pathogen, *H. pylori* [367], and demonstrates the biocompatibility of *Bifidobacterium* for further clinical application.

#### 4.1.4. *Bifidobacterium* Cell Surface

Several strains of *Bifidobacterium* have been investigated for various modalities of cancer treatment, but of specific interest for the scope of this review is *B. breve*, with characteristic expression of the cell surface localized exopolysaccharide (EPS) with a known immunomodulatory role (Figure 7B) [365,365]. Expression of EPS forms has been shown to allow *B. breve* to colonize and persist in mice through reduction in proinflammatory cytokine levels, thereby impairing both innate and adaptive cell responses [365]. Immune cells interact with EPS via TLR-4, resulting in inhibition of MAPK and NF-κB pathways to attenuate the host immune response (Figure 2) [370]. EPS may be able “mask” other surface antigens, allowing for evasion of antibody recognition. The polysaccharide immunogens of EPS evaded the complement cascade and immune cells, decreasing cytokine stimulation to prevent a humoral immune response [365]. In mouse models, a EPS^+^ *B. breve* strain did not result in a detectable host immune response, but the EPS^−^ strain elicited a strong response and was quickly cleared [365]. Further, serum antibody titers from the EPS^−^ *B. breve* demonstrated stronger antigen-specific total Ig response [365], while subjects exposed to the EPS^+^ strain had significantly increased IFN-γ, TNF-α, and IL-12^+^ T and B cells.

It has been determined that *EPS* gene clusters are highly variable in genetic composition among *Bifidobacterium* species [371]. The functional diversity of this variability results in strain-specific attributes such as the antimicrobial activity of *B. longum* BCRC 1464 to defend against pathogens, or the ability of *B. longum* subsp. *longum* 35,624 (previously *B. longum* subsp. *infantalis*) to stimulate T regulatory cells in the human intestine and increase circulating Fox3^+^ lymphocytes, thus preventing inflammation [371]. It is worth noting that horizontal gene transfer is likely the original *B. breve* acquisition source for the *EPS* gene [365], raising important questions regarding further horizontal gene transfer to other species. Targeted synthesis of EPS on the surface of other oncolytic bacterial species, however, may give rise to phagocytosis evasion and thus allow the modified microbes to travel to the tumor at higher rates.

#### 4.1.5. *Bifidobacterium* Metabolism and Byproduct Secretion

In general, fermenters, use a “bifid shunt” pathway to metabolize hexose [363]. In the presence of inorganic phosphate, this saccharolytic pathway degrades fructose-6-phosphoketolase into fructose-6-phosphate [363].

#### 4.1.6. *Bifidobacterium* Host Interaction

When intravenously treated with a cocktail of *B. bifidum*, *B. longum*, *B. lactis*, and *B. breve* in tandem with an anti-CD47 immunotherapy in mouse models of colon adenocarcinoma (MC38) and T cell lymphoma (EG7) cell lines, *Bifidobacterium* localized within the tumor environment and utilized the STING (stimulator interferon genes) pathway to regulate type I IFN signaling. This resulted in elevated IFN-β and promotion of cross-priming in tumor dendritic cells [372], enabling both innate and adaptive antitumor responses within the tumor microenvironment [372]. Elucidating mechanistic details of this pathway will likely extend to modification of other oncolytic species to result in similarly improved antitumor effects.

In a landmark study for oncolytic bacteria, the engineered *B. longum* strain AOS001F/5-FC strain carries a plasmid that demonstrated increased cytosine deaminase activity, resulting in generation and accumulation of the chemotherapeutic 5-fluorouracil (5-FU) specifically within the tumor microenvironment (Figure 7B) [373]. In colorectal cancer (CT26) BALB/c models, combination of the AOS001F/5-FC with anti-PD-1 therapy resulted in both an elevated CD4^+^ T cells at the tumor site and an increased the CD8/T reg ratio [373]. The combination of these effects resulted in suppression of tumor development and ultimately longer subject survival [373]. Notably, rejection of a subsequent tumor challenge was observed when mice were inoculated with dose of tumor cells five times larger than the initial implantation, demonstrating that adaptive anti-tumor memory was successfully developed [373]. Without combination with anti-PD-1 treatment, engineered *B. longum* strain APS001F/5-FC was able to suppress tumor growth; however, without an overall increase observed in survival time [373]. This direct delivery of treatment penetrating into the harsh microenvironment of a solid tumor gives *Bifidobacterium* a stark advantage over current therapeutics. Further investigation is warranted to identify what is causing this modified *B. longum* to induce a memory T cell response specific to the CT26 tumor cells. Elucidating the details of this mechanism will help identify which molecules are associated with the antitumor memory immune response so that they can be harnessed for further clinical application.

#### 4.1.7. *Bifidobacterium* Oncolytic Development

In addition to prophylactic capacities, *Bifidobacterium* strains have undergone engineering to deliver therapeutic agents directly to the tumor core, elegantly navigating around the difficulty of canonical drug delivery in this harsh microenvironment. *B. infantis* was linked to a nanoparticle carrier of adriamycin (DOX), a chemotherapeutic agent [368]. Because bacteria are innately inclined to forage for proteins, the DOX-NPs directly bound to the *B. infantis* cell surface when incubated together in suspension, creating a biohybrid known as Bif@DOX-NP [368]. Solid tumors expressing Secreted Protein Acidic and Rich in Cysteine, or SPARC, have a natural affinity for binding albumin present in composition of the biohybrid chemotherapeutic Bif@DOX-NP (Figure 7B). This affinity was exploited to enhance tumor specific delivery through binding of gp60 receptors on the surface of tumorigenic cells to initiate the release of DOX-nanoparticles within the tumor [368]. After intravenous administration in a mammary carcinoma (4T1) BALB/c models, this biohybrid-conjugated treatment modality resulted in higher intra-tumoral DOX levels when compared to treatments of *B. infantis* alone or DOX-NPs without *B. infantis* [368]. Overall, Bif@DOX-NP treatment slowed tumor growth, but also inhibited tumor cell migration, as well as tumor regression, indicating the potential and flexibility inherent to this type of oncotherapeutic [368]. This methodology is worth imitating in other oncolytic bacteria because of its ability to harness bacteria’s inclination to directly bind protein-coated therapeutics which, in turn, interact with tumor cells of the various cancers that feature SPARC expression.

Similarly, *B. longum* has been engineered to carry the sonosensitizer, hematoporphyrin monomethyl ether (HMME) (Figure 7B). In this study, a simple electrostatic charge was used to bind cationic polyethyleneimine (PEI) to the negatively charged phosphate and carboxyl groups naturally present on the *B. longum* cell surface, thus adding reactive primary amines [369]. The PEI was then conjugated to the carboxyl groups on the HMME to complete the final engineered *Bifidobacterium* cells, HMME@Bif [369]. Colorectal cancer (CT26) BALB/c mouse models were treated with intravenous HMME@Bif and SR717, a STING-agonist, and then ultrasound irradiation was used to activate the sono-sensitive HMME, thus generating reactive oxygen species thought to be responsible for the destruction of tumor cells. This destruction releases tumor antigens and increases concentrations of IL-6, TNF-α, and IFN-γ, stimulating dendritic cell maturation, NK cell activation, and T cell response promotion [369]. This oncolytic bacteria-mediated therapeutic modality succeeded in activating the host antitumor immune response, ultimately inhibiting both primary and metastatic tumors. This methodology of engineering bacteria as a sonosensitizer carrier takes advantage of the inherently negatively charged components of bacteria cell walls to bind the HMME. Further, applying ultrasound to activate the accumulated HMME delivered by the bacteria provides a minimally invasive modality to physically obliterate solid tumors. When relying on electrostatic charge to engineer bacteria as a sonosensitizer carrier, certain factors should be considered to reduce the potential for inadvertently disrupting the interaction between the charged groups, such as the surrounding pH or the type of substances within the treatment solvent or in vivo.

### 4.2. Caulobacter

#### 4.2.1. *Caulobacter* Basic Microbiology

*Caulobacter* is a Gram-negative aerobe with a polar flagellum or stalk giving rise to motility (Figure 7A) [374]. These curved rod shaped bacteria have no catalase activity [374].

#### 4.2.2. *Caulobacter* Genome

The median genome length reported for *Caulobacter* is 4.25 Mbp, with a median GC content of 67.9% [375].

#### 4.2.3. *Caulobacter* Background and History

*Caulobacter* is a stalked, Gram-negative bacteria naturally found in soil and aquatic environments, including tap water, rivers, and freshwater lakes [374,376,377,378]. Under wet mounts, the cells present in a rosette formation [374]. Interestingly, *Caulobacter* undergoes asymmetric cell division in which the daughter cell has a polar flagella that is eventually lost, and a stalk develops in its place [374]. While is not typically found as part of the human microbiota, it is considered non-pathogenic [376,377,378]. *Caulobacter* is determined to be a generally safe option for biotherapy development because it cannot proliferate within a human host [376] and a unique lipid A structure is much less immunogenic than the LPS of enteric bacteria [378].

#### 4.2.4. *Caulobacter* Cell Surface

*Caulobacter* has an S-layer, composed of RsaA monomers organized in a hexagonal pattern on the cell surface (Figure 7B) [377]. The S-layer forms a crystalline coating of protein anchored to the cell surface by lipopolysaccharide (LPS) [376]. Electroporation was employed to introduce plasmids into the *C. crescentus* genome capable of expressing foreign peptides on the S-layer without disturbing function [377]. For example, the *C. crescentus* JS4022 strain was elegantly engineered (Figure 7B) [377] to express and display both protein G and CD4 in the S-layer [377]. Ultimately, antibody capture was not affected, and HIV neutralization was enhanced by CD4 binding [377]. The success of engineering the S-layer to display foreign peptides that maintain function makes *C. crescentus* a promising option for further prophylactic development as there is profound potential to express and display chemotherapeutic agents or other tumor-specific antibodies for targeted, specific delivery.

Another structure unique to the *C. crescentus* cell surface is the lipid A within the outer membrane LPS (Figure 7B) [378]. Lipid A lacks phosphates, and has an unusual DAG backbone as well as uncommon fatty acids [378]. *C. crescentus* has been shown to be much less toxic than other Gram-negative bacteria with a canonically highly immunogenic LPS, which is thought to be due to this unique lipid A endotoxin [378]. Lipid A endotoxin activity demonstrated a significantly weaker immunogenicity level in vitro than a recombinant LPS isolated from *E. coli* [378]. Further studies have demonstrated even very high doses of *Caulobacter* fail to induce a severe inflammatory response [378]. Notably, even with this characteristically low toxicity, *Caulobacter* remains capable of stimulating an innate immune response, supporting its potential as an adjuvant [378]. LPS has been shown to activate the complement system to increase C5a as well as interact with immune cells via TLR-4 [19,26], both stimulating the NF-κB pathway [379] to release inflammatory cytokines such as IL-1, IL-6, IL-8, TNF-α, and IFN-γ, and to recruit immune cells for an adaptive immune response (Figure 2). Safety is a crucial aspect in developing oncolytic bacteria for clinical application, which, particularly when paired with the homogenous nature of the S-layer, makes *C. crescentus* an excellent candidate for engineering to generate further prophylactic capacities.

It must be mentioned that a single study isolated a hospital-acquired strain of *Caulobacter* from immunocompromised patients who had developed infections and compared it to environmental *Caulobacter* strains [380]. The clinical isolate, *C. mirare*, was and compared with environmental *C. crescentus* and *C. segnis* to assess potential for virulence using *Galleria mellonella*, the greater wax moth—a model used for evaluating infection potential of bacteria due to this host’s ability to visibly produce melanin that can be monitored as the immune system responds [380]. Quantitative evaluation of health span assays determined the clinical and environmental species of *Caulobacter* had similar degrees of pathogenicity inducing an immune response in *G. mellonella*, compared to no immune response when injected with nonpathogenic *E. coli* MG1655 [380]. The results demonstrated that both the clinical and environmental strains shared a feature that allowed them to infect the host [380]. While no strains have had pathogenicity islands, virulence factors, or host evasion mechanisms observed, the LPS composition of *Caulobacter* can result in opportunistic infections, particularly in immunocompromised patients [380]. Studies in other models with higher translatability for human research have shown *Caulobacter* to be safely utilized in oncotherapy research [376,377], but the immunocompromised status of patients cannot be ignored when evaluating to which patient populations a *Caulobacter*-augmented treatment should be administered. Of note, infection that may arise from *Caulobacter*-mediated oncotherapy is easily resolved with antibiotics and have a favorable profile of side-effects in preliminary studies, especially compared to many current therapeutics.

#### 4.2.5. *Caulobacter* Metabolism and Byproduct Secretion

*C. crescentus* secretes bacteriocin-like proteins, CdzC and CdzD, capable of toxicity to cells lacking Cdz immunity (Figure 7B) [381]. The contact-dependent inhibition by glycine zipper proteins (Cdz) system secretes these proteins via a type I secretion system [381]. However, CdzC and CdzD accumulate on the surface as cell-associated aggregates rather than following conventional extracellular secretion of bacteriocins [381]. Both proteins are required because they require direct contact to form pores in the inner membrane of a neighboring cell, disrupting its cell envelope integrity [381]. The Cdz system is upregulated during stationary phase as a mechanism for intracellular competition when nutrients are scarce, and can be important for outcompeting pathogens [381].

#### 4.2.6. *Caulobacter* Oncolytic Development

The antitumor activity of *C. crescentus* was probed in murine models of lung cancer, breast cancer, and leukemia [376]. These studies unanimously resulted in longer survival and decreased tumor size, indicating efficacy across multiple types of solid tumors [376]. *C. crescentus* was unable to proliferate in vivo, and was cleared from the subjects by 10 days post-injection [376]. Intriguing, *C. crescentus* colonization within the tumors was not observed, seeming to indicate the antitumor effects were generated by the host immune response thought to be activated by the LPS-anchored S-layer [376]. Another study seems to confirm *C. crescentus*-presented LPS initiates host immune response, recording evidence of NK cell stimulated release of IFN-γ [382]. Intraperitoneal injection of *C. crescentus* in breast cancer mouse models resulted in smaller tumors as well as extended survival [382].

Intraperitoneal injections of *C. crescentus* were administered before and after subcutaneous tumor implantation (EL4 or B16 cells) in C57BL/6 mice, Jα18^−/−^ mice (lacking type I NKT cells), and CD1d^−/−^ mice (lacking CD1d-restricted NKT cells) [383]. *C. crescentus* treatment in WT mice significantly slowed EL4 tumor growth compared to PBS controls but had no observable impact on B16 tumor growth. B16 cells are known to lack expression of CD1d [383]. There was also no tumor growth inhibition observed in CD1d^−/−^ mice, indicating that NKT cells are necessary for controlling tumor development [383]. In Jα18^−/−^ mice, *C. crescentus* treatment significantly slowed both EL4 and B16 tumor growth compared to WT, suggesting type I NKT cells have an immunomodulatory role against CD1d-restricted NKT cells that prevents antitumor activity [383]. When Jα18^−/−^ bone marrow derived DCs were plated in vitro with *C. crescentus*, lower levels of IL-10 and higher levels of IL-12p70 were produced. IL-12p70 is a cytokine known to shift T cells towards a Th1 phenotype and is also thought to be involved in antitumor immunity [383]. CD40, CD86, CD80, and IL-12 expressions were also elevated, suggesting these DCs stimulated by *C. crescentus* may enhance activation of antigen-specific T cells and NK cells [383]. While largely considered non-pathogenic, *C. crescentus* is capable of instigating a strong T cell response [383]. Therefore, it is proposed that *C. crescentus* triggers the innate immune response, stimulates CD1d recognition and presentation of tumor associated antigens by DCs, activating NKT cells to produce IFN and resulting in immunosurveillance that responds to tumor cells expressing CD1d [383]. Further research to identify what specific molecules or antigens of *C. crescentus* are responsible for initiating the type II NKT cell immune response would be of great value to harness and improve antitumor effects.

### 4.3. Lactobacilli

#### 4.3.1. *Lactobacilli* Basic Microbiology

*Lactobacilli* are Gram-positive Facultative anaerobic that do not form spores (Figure 7A) [384]. These rod shape bacteria are catalase negative [384].

#### 4.3.2. *Lactobacilli* Genome

The median length reported for *Lactobacilli* is 1.76 Mbp, with a median GC content of 38.8% (Figure 7A) [385].

#### 4.3.3. *Lactobacilli* Background and History

*Lactobacilli* are commonly found inhabiting the human microbiota [386,387] in the mouth, gastrointestinal tract, and vagina [384]. *Lactobacilli* are critical for the production of many dairy products (e.g., cheese, yogurt), fermented foods (e.g., pickles, olives) [384] and supplements because of its probiotic capabilities [386]. Studies have investigated the effects both as a prophylactic capable of preventing cancer development and as a treatment for cancer. The antitumor effects of *Lactobacillus* have made it a topic of particular interest for its potential as an immunoadjuvant or drug delivery system.

#### 4.3.4. *Lactobacilli* Cell Surface

Some *Lactobacillus* strains, such as *L. acidophilus*, have been characterized to express lipoteichoic acid (LTA) in the cell surface [387] capable of triggering cytokine release and dendritic cell stimulation (Figure 7B) [387]. LTA has been shown to interact with immune cells via TLR-2 [22,26,387] and also activates the complement system to increase C5a, [26] promoting the NF-kB pathway [379] to release pro-inflammatory cytokines and inducing an enhanced host immune response [26,379,387] (Figure 2). Colon polyp (TS4Cre × APC^lox468^) murine models treated with the *L. acidophilus* LTA^+^ strain, NCK56, exhibited an immune response. In contrast, those given the LTA^−^ strain, NCK2025, had levels of intrapolyp mast cells, reduced T regs, and decreased dendritic cells levels like those of healthy mice. Cumulatively, treatment with the NCK2025 LTA^−^ strain downregulated inflammation, resulting in a “reset” immune environment that protected against the potential development of colon cancer [387]. Taking advantage of LTA’s ability to regulate the host immune system could be a simple tool to exploit the characteristic immunogenicity of LTA to attract and target the immune response. Conversely, removal of LTA, could allow *Lactobacillus* or other LTA-expressing bacteria localized more efficiently without stimulating an immune response.

Another protein found on many species of *Lactobacillus* is the S-protein, which is the main component of the cell surface S-layer (Figure 7B) [388]. The S-layer allows *Lactobacillus* to strongly adhere to enterocytes, providing protection in the form of a physical barrier over the cells of the intestine and inhibiting pathogen adherence [388]. Strains of *L. salivarius*, *L. reuteri*, and *L. johnsonii* have undergone in vitro evaluation with intestinal epithelial cell (Caco-2) cultures. Some strains, such as *L. reuteri* JN981858 and JN981, indicated high levels of adhesive ability [388]. In contrast, *L. salivarius* ZJ614, *L. reuteri* ZJ616, *L. reuteri* ZJ617, *L. reuteri* ZJ621, and *L. reuteri* ZJ616 were all strong inhibitors of *E. coli* K88 and *S. enteritidis* 50,335 adhesion to intestinal epithelial cells. Removal of the S-layer reduced this ability [388]. These species-specific nuances indicate other factors likely influence these interactions.

#### 4.3.5. *Lactobacilli* Metabolism and Byproduct Secretion

Optimal growth of *Lactobaccili* generally occurs under microaerophilic conditions and because the primarily are carbohydrate fermenters, produces lactic acid [384]. A metabolite isolated from *L. acidophilus*, cb-EPS, has indicated antitumor effects (Figure 7B). In human colorectal cancer (HT-29) cell cultures, cb-EPS treatment significantly affected cell morphology, but intriguingly not the cell cycle, and showed both dose-dependent and temporal-dependent action inhibiting cell proliferation [389]. cb-EPS may induce tumor cell autophagy via regulation apoptotic factors since Bcl-2 expression decreased but Bak increased [389]. Beclin-1, an inducer of autophagy factor expression, and GRP78, a protein involved in cell processes such as endoplasmic reticulum stress, both had increased expression after cb-EPS treatment, indicating that cb-EPS may regulate these signaling molecules through mechanisms of ER stress to ultimately promote autophagy in cancer cells [389]. *L. acidophilus* produced cb-EPS has the potential to be harnessed as an onco-therapeutic agent, but in vivo studies are needed for further investigation.

#### 4.3.6. *Lactobacilli* Oncolytic Development

*L. casei* has been studied for its prophylactic effects in BALB/c models [390,391]. When administered orally prior to the introduction of colon carcinoma (CT26) tumor cells, *L. casei* treatment resulted in persistent elevated IFN-γ and IL-12 within the TME, attracting cytotoxic T cells and NK cells to the tumor. Ultimately, this treatment could inhibit tumor growth by stimulating Th1 immune response and promoting cytotoxic T cell migration to the tumor site [390], generating an immunomodulatory effect with notable potential for further development as an oncotherapeutic adjuvant.

Different routes of administration of *L. casei* have also been studied, with intriguingly different outcomes. In lung cancer (TC-1) C57BL/6 models, intranasal administration resulted in reduced tumor onset [386]. When administered subcutaneously, *L. casei* reduced tumor size, but not protect against tumor onset [386]. Both subcutaneous and intranasal administration of *L. casei* demonstrated increased IL-2 levels—recruiting lymphocytes and NK cells to initiate the host anti-tumor response [386]. The intranasal treatment showed a negative correlation between tumor size and CD3^+^, CD8^+^, NK cell abundance, and Foxp3^+^, indicating a systemic immune response [386]. Understanding and determining the best route of administering *L. casei*, and oncolytic bacterial in general, will prove critical for clinical translation.

The specific migration and accumulation of *L. casei* to the tumor microenvironment has also been studied in ddY mice with sarcoma (S-180) cells and BALB/c mice with colon adenocarcinoma (C26) cells [391]. In this case, vascular mediators were administered prior to treatment with *L. casei* to enhance delivery through leaky vasculature characteristic of tumors [391]. Nitroglycerin, a common and well-studied medication, significantly increased bacteria delivery, and improved the antitumor effects of *L. casei* resulting in a prolonged survival rate [391]. Enalapril, an ACE inhibitor, also increased delivery and accumulation of the bacteria, but interestingly, not as drastically [391]. After intravenous administration, *L. casei* was quickly cleared from normal tissue but accumulated specifically within tumor tissue [391]. Levels of Inflammatory cytokines such as IL-6, MCP-1, and TNF-α were altered in serum, but increased TNF-α and NOS in tumor tissues was observed, even more so when augmented with nitroglycerin [391]. This study demonstrates the potential for *L. casei* as a carrier for targeted drug delivery to tumor cells without harmful or pathogenic effects on other tissues. Further, the application of vascular mediators to increase tumor permeability and enhance delivery of bacteria to the tumor site is not specific for use with *L. casei* and could be used with other anaerobic bacteria as well. 

## 5. Future Perspective

Oncolytic bacteria are a rapidly progressing and expanding field of study that is just beginning to see clinical translation. The oncolytic capabilities of these bacteria stem from surface proteins, secreted proteins, and metabolic capabilities. A greater understanding of these characteristics should be at the forefront of current research to better comprehend anti-cancer capacities and immune modulation. While usually incredibly selective and specific, it is possible for these bacteria to secrete harmful toxins or trigger immune responses that are destructive to normal, healthy cells. Genetic engineering and directed evolution can be used to not only address safety issues, but to expand the oncolytic capability of bacteria and improve effectiveness in different types of tumors through harnessing unique components.

Significant challenges face these developing fields of oncolytic and prophylactic bacteria in order to ensure their safety and reproducibility. Additionally, both the cost and availability of these bacteria should be considered. Given the intrinsic capacity for replication, live biologic therapeutics could serve to address socioeconomic issues intrinsic to health care. This field cannot forget how the dysregulation and improper implementation of Coley’s toxin led to both negative patient outcomes and ultimately widespread mistrust from patients, clinicians, and the general public. Current studies must prove these bacteria can not only be effective, implemented safely and with widespread reproducibility. Careful, well-verified modification of these oncolytic and prophylactic bacteria can and will lead to treatment options capable of ‘hijacking’ the host immune response, harnessing it to mitigate solid tumors.

## 6. Conclusions

Oncolytic bacteria offer many promising avenues for novel cancer therapeutics by providing a more targeted, effective approach for either bacterial-mediated anti-cancer activities or bacterial-mediated drug delivery. Recent advances in bioengineering, directed evolution, and synthetic biology have allowed for the amplification and/or addition of further anti-cancer capacities intrinsic to these bacterial species, even extending as far as prophylactic anti-cancer treatments preventing the formation of solid tumors. Solid tumors provide unique microenvironments that while challenging to canonical therapeutics, represent an advantageous physiological niche for the colonization and subsequent lysis of cancer cells by these oncolytic bacteria. While surgery, radiation, and chemotherapy remain mainstays for treating solid tumors, oncolytic bacteria offer several potent advantages over these modalities—including the potential to overcome drug resistance and reduce the risk of tumor recurrence. The creativity, flexibility, and innovation demonstrated by these growing fields are encouraging, lending to the belief that the question of if cancer can be cured will soon shift to when it will be cured. Technological advancement and the use of oncolytic bacteria development is expected to herald a new era in pharmaceutical research and development of cancer treatments. 

## Figures and Tables

**Figure 1 pharmaceutics-15-02004-f001:**
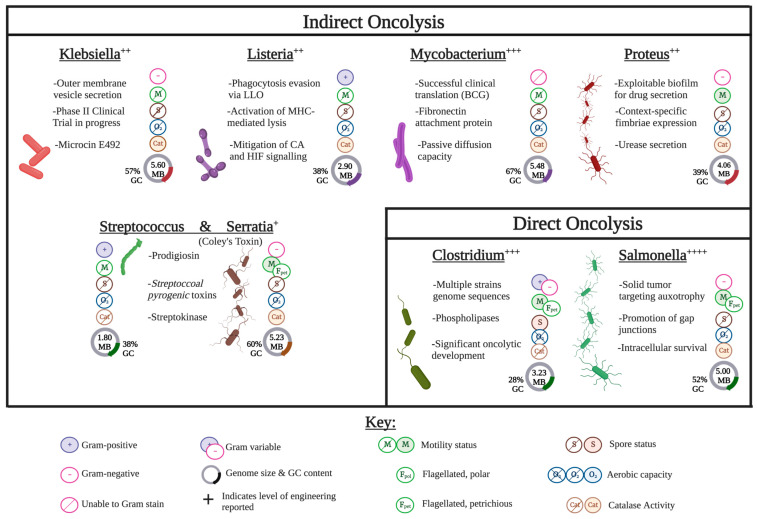
Promising attributes of oncolytic bacteria under development toward clinical translation for consideration to add further anti-cancer capacities.

**Figure 2 pharmaceutics-15-02004-f002:**
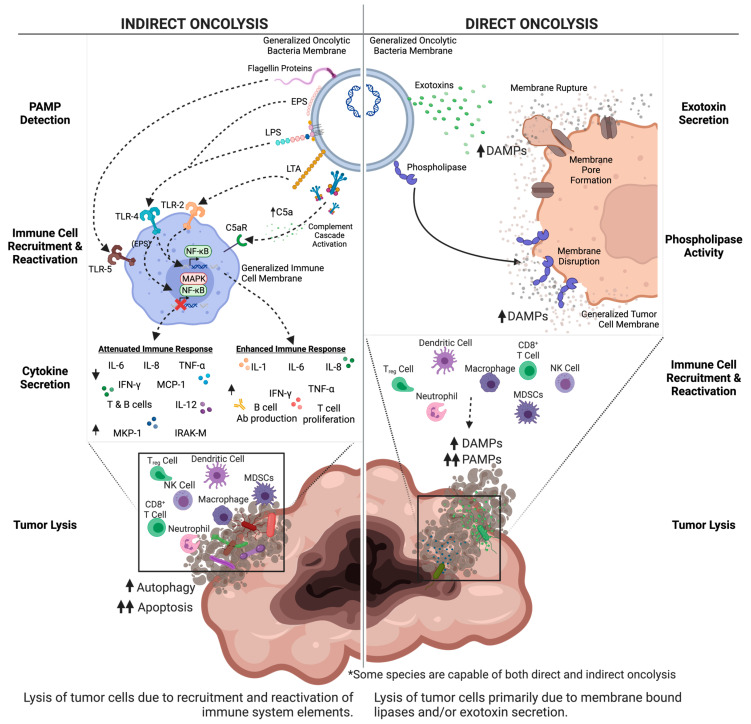
Generalized bacterial membrane components known to contribute to oncolytic properties and the host immune response resulting in oncolytic effects. Dashed arrows represent indirect, subsequent effects, solid arrows direct effects. * Indicates some species are capable of both direct and indirect oncolysis and thus can activate multiple pathways depicted.

**Figure 3 pharmaceutics-15-02004-f003:**
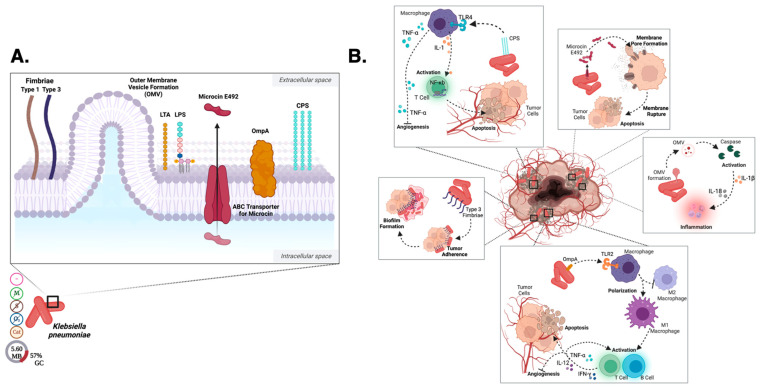
*Klebsiella pneumoniae* (**A**) membrane components and proteins contributing to oncolytic properties and (**B**) host immune response resulting in indirect oncolysis.

**Figure 4 pharmaceutics-15-02004-f004:**
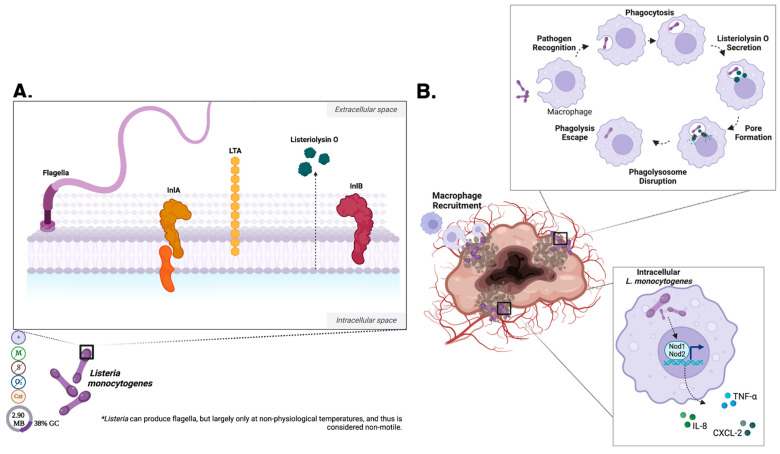
*Listeria monocytogens* (**A**) membrane components and proteins contributing to oncolytic properties and (**B**) host immune response resulting in indirect oncolysis. * Indicates *Listeria* can produce flagella, but largely only at non-physiological temperatures, and thus is considered non-motile.

**Figure 5 pharmaceutics-15-02004-f005:**
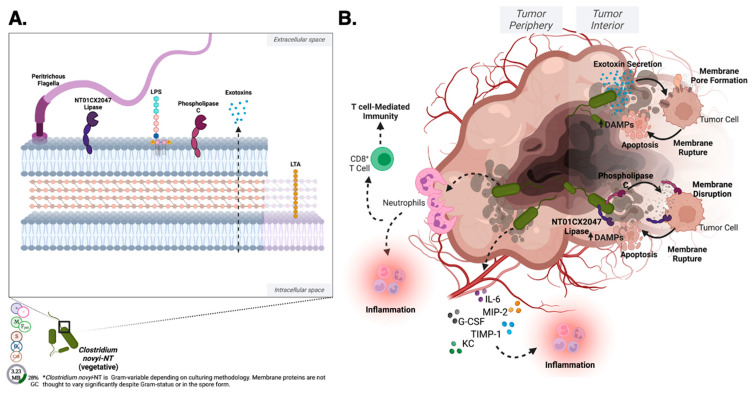
*Clostridium novyi*-Non Toxic (NT) (**A**) membrane components and proteins contributing to oncolytic properties and (**B**) host immune response resulting in direct and indirect oncolysis. Dashed arrows represent indirect, subsequent effects, solid arrows direct effects. * Indicates *C. novyi*-NT is Gram-variable depending on culturing methodology. Membrane proteins are not thought to vary significantly despite Gram-status or in the spore form.

**Figure 6 pharmaceutics-15-02004-f006:**
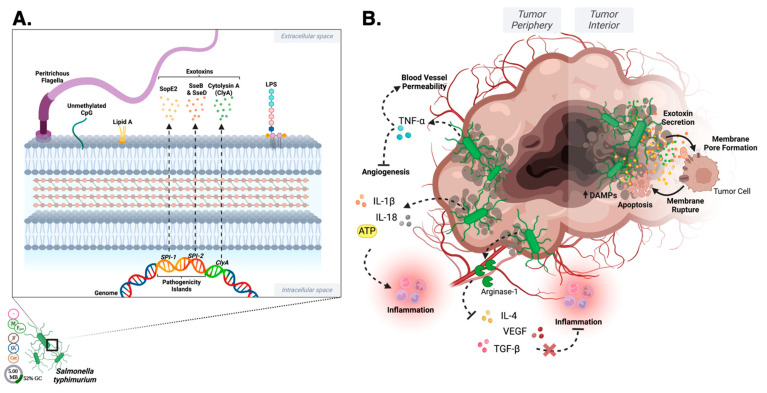
*Salmonella typhimurium* (**A**) membrane components and proteins contributing to oncolytic properties and (**B**) host immune response resulting in direct and indirect oncolysis. Dashed arrows represent indirect, subsequent effects, solid arrows direct effects.

**Figure 7 pharmaceutics-15-02004-f007:**
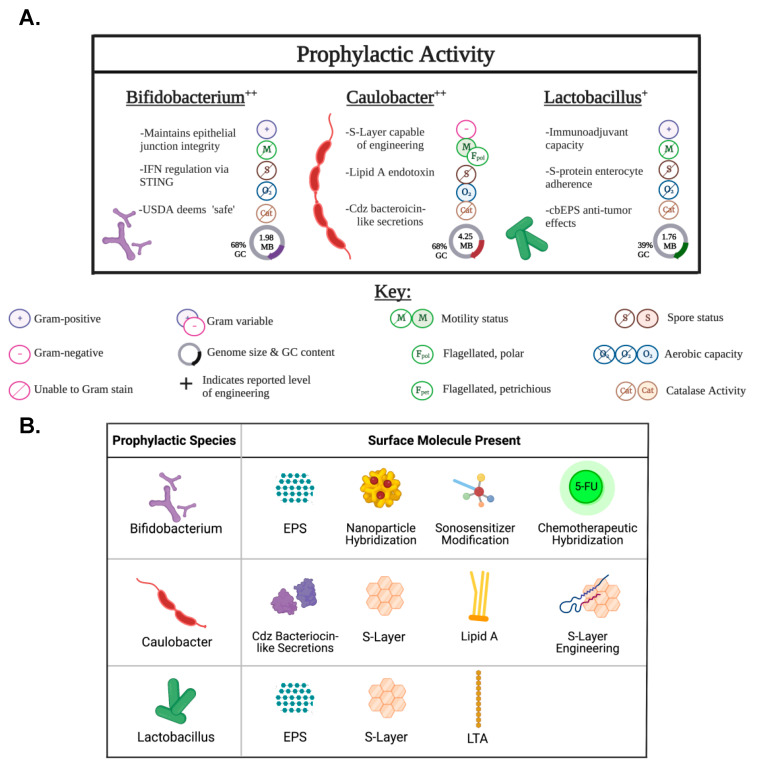
Prophylactic bacteria (**A**) general attributes of species under development and (**B**) Comparison of membrane components and proteins contributing to anti-cancer properties of interest.

## Data Availability

Not applicable.

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
