# Peer review of "Hacking the Immune Response to Solid Tumors: Harnessing the Anti-Cancer Capacities of Oncolytic Bacteria"

_pharmaceutics, 2023, doi:10.3390/pharmaceutics15072004_

Round 1

Reviewer 1 Report

This manuscript systematically summarized the oncolytic bacteria and their particularly favorable characteristics, which contribute to understanding of the bacteria-host interaction, assessing anti-cancer capacities and unlocking the full cancer therapeutic potential of oncolytic bacteria. But I have some following concerns: 

  1. The potential side effects of oncolytic bacteria (Klebsiella, Listeria, Mycobacteria, Streptococcus/Serratia (Coley’s Toxin), Proteus, Salmonella, and Clostridium Klebsiella, Listeria, Mycobacteria, Streptococcus/Serratia (Coley’s Toxin), Proteus, Salmonella, and Clostridium) should also be discussed because some of these bacteria are potential human pathogens. 
  2. Some tables should be summarized and added to the manuscript in order to systematically and comprehensively show the readers the species, mechanism, dosage, efficacy, side effects and other information of direct and indirect oncolytic bacteria. 
  3. It is best to indicate the source of the picture or apply for copyright to avoid potential property disputes. 
  4. Please unify the format of references in the text, including the author's abbreviation, the case of words in the title, and the italics of species names. 
    In a word, I recommend major revision for this manuscript. 

Author Response

We are greatly encouraged that Reviewer 1 along with the other reviewers sees the potential our review has to impact the development oncolytic bacterial as oncotherapeutics through its original perspective regarding harnessing the innate mechanisms of immunomodulation, subsequently generating bacterial-mediated therapeutics, and furthering the field of biologic therapeutics in general. We thank the reviewer for the time they clearly spent carefully critiquing of this manuscript to make it a stronger overall study. Each comment has been carefully considered by our team with revisions made where applicable. We would like to directly address the following comments:

  1. Other thoroughly detailed reviews are available (for example refs 1, 57, 63, 304, 313, 315, 318 among others) that outline the clinical pathogenic effects of the bacteria discussed in the current review. This current manuscript focuses instead on the mechanisms of action (e.g., cytokine response) underlying these pathogenic effects. We have attempted to detail these in the context of current synthetic biology advances such that the immune response might be harnessed and manipulated for added anti-cancer efficacy.
  2. The details of species, mechanism, dosage, efficacy, side effects and other information of direct and indirect oncolytic bacteria have been thoroughly summarized elsewhere in the literature, for example in references 1, 57, 63, 304, 313, 315, 318 among others. Considering the availability of this data in multiple formats, including tables, we choose to refer readers to other relevant reviews rather than reiterate already published material.
  3. All the figures included in this manuscript are original works created by this authorship team using Biorender with appropriate licenses for this review and have never been published before. Citations for the studies summarized by these figures are thoroughly detailed in the corresponding body text section as per Pharmaceutics formatting guidelines.
  4. Thank you for pointing this out. We will make sure all references are in line with Pharmaceutics requirements prior to publication once the manuscript of the text is set.

Reviewer 2 Report

The manuscript “Hacking the Immune Response to Solid Tumors: Harnessing the Anti-Cancer Capacities of Oncolytic Bacteria” is a comprehensive review of the usage of bacteria in cancer treatment of solid tumors. It is not the first review about oncolytic bacteria and at least one of its authors - K.Dailey – is among co-authors of at least two similar reviews (references ##54-55). So, the current manuscript continues the line and looks like the authors – most from one research center – are in the start of development of new oncolytic strains of different types of bacteria.  The current manuscript is quite useful and contains some new information in comparison with previous reviews, therefore its publication would be useful for scientists who are working in cancer treatment development area.

The shortages of this review are:

1.       The two above mentioned reviews are not cited at the Introduction and Background chapter. And there is no explanation in the same chapter what are the differences between the reviews 54-55 and the current manuscript. And there are no information what is new in this review.

2.       The references to 54-55 papers are presented in the form of primary sources of information in spite of the fact that these papers are the reviews and not the experimental publications.

Author Response

We are greatly encouraged that Reviewer 2 along with the other reviewers sees the potential our manuscript has to impact the development oncolytic bacterial as oncotherapeutics through its original perspective regarding harnessing the innate mechanisms of immunomodulation. This manuscript subsequently has the potential to inspire bacterial-mediated therapeutic development, furthering the field of biologic therapeutics. We thank the reviewer for the time they clearly spent carefully critiquing this manuscript to make it a stronger overall study. Each comment has been carefully considered by our team with revisions made where applicable. We would like to directly address the following comments:

  1. We have revisited the Introduction and Background section to add the following statement of purpose for clarification “The purpose of this manuscript is to provide a unique, unpublished perspective in that it specifically pairs literature detailing oncolytic bacteria surface characterization with the reported host-immune bidirectional interactions in pathogenic contexts to develop novel oncotherapeutics and live biologic therapies with the ultimate goal of clinical translation.” This purpose is distinct from references 54-55 (now ref 56-57), as well as other available reviews both cited and uncited in the current manuscript in that it’s a logical extension of that work. Not only is this purpose distinct, but the links between surface characterization and host immune response in pathogenic context are also not clearly defined in previous reviews.
  2. We thank the reviewer for indicating we did not clearly communicate references 54-55 (now ref 56-57) are reviews and not primary publications. We have modified the section as follows to provide additional clarity “Ultimately, this creates an environment that is hostile to cancer cells while promoting the activation of immune cells that can attack the tumor as previously reviewed[56]. Clostridium novyi-NT, for example, is theorized to utilize LPS to stimulate the innate immune response[57]. In this context, LPS is hypothesized to trigger the release of pro-inflammatory cytokines, leading to the activation of immune cells and the recruitment of immune cells to the site of the tumor as previously reviewed[57].” and “Both the hurdle and method to overcome this challenge towards clinical translation have been previously reviewed[56] with applications for C. tetani development as well as other oncolytic bacteria.”

Reviewer 3 Report

Suggest to  discuss in more detail clinical trials 

its OK

Author Response

We agree with the reviewer that clinical trial results provide critical context for progressing these novel therapeutics towards clinical translation. However, our previously published review as well as other publications cited as examples in relevant sections provide readers with references to more detail regarding clinical trials. The purpose of this manuscript is to provide a unique, unpublished perspective in that it specifically pairs literature detailing oncolytic bacteria surface characterization with the reported host-immune bidirectional interactions in pathogenic contexts to develop novel oncotherapeutics and live biologic therapies with the ultimate goal of clinical translation. Therefore, a thorough discussion detailing clinical trial results and progress is beyond the scope of the current manuscript.

Reviewer 4 Report

Summary: This article is a very detailed and complete history and review of the field of oncobacteria.

Detailing their history, the components of relevance for oncolysis and the experiment carried out with

these bacteria noting the cellular engineering process there had been carried out and new avenues of

attack. The review dives into the oncolytic bacteria by direct and indirect mechanisms and also makes a

small section for prophylactic bacteria that more than likely are part of the microbiota-immune

interactions.

Comments.

-In the introduction maybe, it will be desirable to remark on some of the therapeutic strategies also

targeting the innate immune system as adjuvant mainly and their effects so far to justify why indirect

oncolytic bacteria are different from current approaches (TLR-4,5,9 antibodies and more)

-Note that when referencing one of the experiments carried out in all cases clarifying the model and the

exact cell line of origin is crucial which was present in almost all cases but there were exceptions, and

the assertion then is made too vague. This clarification is important.

-It may be a little too long to read and maybe some sections could be collapsed into single paragraphs.

NA

Author Response

We are greatly encouraged that Reviewer 4 along with the other reviewers sees the potential our manuscript has to impact the development oncolytic bacterial as oncotherapeutics through its original perspective regarding harnessing the innate mechanisms of immunomodulation. This manuscript subsequently has the potential to inspire bacterial-mediated therapeutic development, furthering the field of biologic therapeutics. We thank the reviewer for the time they clearly spent carefully critiquing this manuscript to make it a stronger overall study. Each comment has been carefully considered by our team with revisions made where applicable. We would like to directly address the following comments:

  • We briefly discussed other biological adjuvants, and have now included further references to reviews regarding other biological adjuvants: “Surgery with adjuvant or neoadjuvant radiation and/or chemotherapy is currently the first line of treatment for solid tumors. Other biologic adjuvants are also known to modulate the immune response by targeting neoplasms as reviewed elsewhere [7,8].” However, the purpose of this review was to demonstrate how the surface molecules on oncolytic bacteria can have a similar effect in addition to their oncotherpeutic capacity.
  • We agree that details regarding the mouse models and cell lines used are critical to correct interpretation of the data reviewed in the current work. We have verified that such information is included with all example experimentation discussed within this manuscript. No exceptions were identified, however we welcome the reviewer’s critic and will be happy to address any specific exceptions identified by the reviewer.
  • We recognize the length of this manuscript is substantial; however, we tried to be very thorough and consistent in the information we present for each oncolytic bacteria species as a guide for the field. In order to best serve a wide audience of readership, we decided not to exclude any species or information that might eventually prove helpful in this developing field of synthetic biology. This manuscript was written in a way that the reader could read sections relevant to their work/interests.

Round 2

Reviewer 1 Report

The authors have addressed all my questions and I recommend accepting this manuscript.

Reviewer 2 Report

The authors presented the corrected version with which I agree.